METHODS AND RESOURCES

# scapGNN: A graph neural network–based framework for active pathway and gene module inference from single-cell multi-omics data

**Xudong Han[1,2], Bing Wang[1,2], Chenghao Situ[2], Yaling Qi[2], Hui Zhu[2]\*, Yan Li[3]\*, Xuejiang Guo[1,2]\***

**1** State Key Laboratory of Reproductive Medicine and Offspring Health, School of Medicine, Southeast University, Nanjing, China, **2** Department of Histology and Embryology, State Key Laboratory of Reproductive Medicine and Offspring Health, Nanjing Medical University, Nanjing, China, **3** Department of Clinical Laboratory, Sir Run Run Hospital, Nanjing Medical University, Nanjing, China

\* njzhuhui@njmu.edu.cn (HZ); yanli@njmu.edu.cn (YL); guo_xuejiang@njmu.edu.cn (XG)

**Data Availability Statement:** All relevant data are within the paper and its Supporting information files. The scapGNN has been implemented as an R package is freely available from CRAN (https://

## Abstract

Although advances in single-cell technologies have enabled the characterization of multiple omics profiles in individual cells, extracting functional and mechanistic insights from such information remains a major challenge. Here, we present scapGNN, a graph neural network (GNN)-based framework that creatively transforms sparse single-cell profile data into the stable gene–cell association network for inferring single-cell pathway activity scores and identifying cell phenotype–associated gene modules from single-cell multi-omics data. Systematic benchmarking demonstrated that scapGNN was more accurate, robust, and scalable than state-of-the-art methods in various downstream single-cell analyses such as cell denoising, batch effect removal, cell clustering, cell trajectory inference, and pathway or gene module identification. scapGNN was developed as a systematic R package that can be flexibly extended and enhanced for existing analysis processes. It provides a new analytical platform for studying single cells at the pathway and network levels.

## Introduction

A biological pathway is a collection of relationships between genes that lead to a certain product or a change in the biological process in the cell [1]. Some databases—such as the Kyoto Encyclopedia of Genes and Genomes (KEGG) and Gene Ontology (GO) databases—have manually grouped interacting or similarly characterized molecules into pathways or gene sets by evidence-supported annotations [2,3]. Biological pathways in distinct cell types have different activation patterns, which facilitates the understanding of cell functions. In single-cell studies, pathway activation analysis has become a powerful approach for the extraction of biologically relevant signatures to uncover the potential mechanisms of cell heterogeneity and dysfunction in human diseases [4,5]. However, the current pathway enrichment analysis methods (e.g., gene set enrichment analysis (GSEA) [6], single-sample gene set enrichment analysis

github.com/XuejiangGuo/scapGNN), GitHub (https://cran.r-project.org/web/packages/scapGNN/index.html), and FigShare (https://figshare.com/articles/software/scapGNN/23734017). The R packages for scapGNN and related scripts are also available from Zenodo (https://doi.org/10.5281/zenodo.8322402).

**Funding:** This work was supported by the National Key R&D Program of China (2021YFC2700200 to XG), the Chinese National Natural Science Foundation (Grants No. 82221005 to XG, 81971439 to XG, 82001611 to YL, 31871164 to HZ, 82071702 to HZ) and the fund from Health Commission of Jiangsu Province (M2020071 to YL). The funders had no role in study design, data collection and analysis, decision to publish, or preparation of the manuscript.

**Competing interests:** The authors have declared that no competing interests exist.

**Abbreviations:** ARI, adjusted rand index; AUC, area under the recovery curve; BCMI, bias-corrected mutual information; COVID-19, coronavirus disease 2019; DEC, definitive endoderm cell; DEG, differentially expressed gene; DNNAE, deep neural network autoencoder; EC, endothelial cell; eP, early pachytene; ESC, embryonic stem cell; FDR, false discovery rate; GAE, GNN autoencoder; GLUE, graph-linked unified embedding; GNN, graph neural network; GO, Gene Ontology; GSEA, gene set enrichment analysis; GSVA, gene set variation analysis; hPSC, human pluripotent stem cell; IAV, influenza A virus; IRS, inner root sheath; KEGG, Kyoto Encyclopedia of Genes and Genomes; LTMG, left truncated mixture Gaussian; NMI, normalized mutual information; PBMC, peripheral blood mononuclear cell; PPAR, peroxisome proliferators–activated receptor; PPI, protein–protein interaction; ROC, receiver-operating characteristic; RWR, random walk with restart; scATAC-seq, single-cell ATAC sequencing; scDART, single-cell Deep learning model for ATAC-Seq and RNA-seq Trajectory integration; scRNA-seq, single-cell RNA sequencing; ssGSEA, single-sample gene set enrichment analysis; SW, silhouette width; TAC, transit-amplifying cell; TRS, transcriptional regulation state; tSNE, t-distributed stochastic neighbor embedding; UMAP, Uniform Manifold Approximation and Projection; VGAE, variational graph autoencoder.

(ssGSEA) [7], and gene set variation analysis (GSVA) [8]) developed for bulk RNA-seq data have been reported to be inappropriate for single-cell sequencing data [9,10]. Compared with RNA-seq data obtained from bulk cell populations, single-cell sequencing data are much sparser, noisier, and lower in library size due to the particular sequencing techniques and experiment protocols [11]. These seriously compromise the accuracy and integrity of gene-level analyses in single-cell data [1]. Hence, an efficient method is urgently needed to parcel out the pathway activity of individual cells.

Recently, some pathway enrichment methods using single-cell RNA sequencing (scRNA-seq) data, such as AUCell [12], Pagoda2 [13], and UniPath [14], have been proposed to study cellular heterogeneity. For example, AUCell calculates the area under the recovery curve (AUC) score for the pathway in the ranked list of genes for each cell as the pathway activity score. Pagoda2 fits a model to renormalize gene expression profiles and uses the first weighted principal component to quantify pathway activity scores. UniPath models the distribution of gene expression as bimodal and converts nonzero expressions into $p$-values. It combines the $p$-values of genes in the pathway and adjusts them as pathway enrichment scores using a common null background model.

These methods still have limitations that make it difficult to mine information from single-cell data. AUCell depends on the ranked list of genes, which allows it to identify only a few pathways associated with top genes at a time. Pagoda2 only focuses on the first principal component, leading to data loss. UniPath needs to construct the null background model for different species, affecting the scalability of the method. Meanwhile, the completeness of the null background model directly affects its performance. Furthermore, these methods do not make inferences about genes with dropout events for the scRNA-seq data with many zero values [9]. Besides, AUCell and Pagoda2 are designed to perform pathway analysis only for single-cell transcriptome data. UniPath proposes a corresponding pathway enrichment method that uses the hypergeometric or binomial test for single-cell ATAC sequencing (scATAC-seq) data, although it still relies on the background distribution.

Moreover, these methods can only be based on predefined pathways or gene sets. Gene modules, serving as building blocks of complex biological networks, are structural subnetworks that exhibit the same organizational patterns or functions [15,16]. Module-based analyses can achieve a higher-level understanding of the design and organization of biological systems. Genomap is an entropy-based cartography method to contrive the high-dimensional single-cell gene expression data into a configured image format and discover cell-specific gene sets [17]. It can compute cell type–specific gene importance scores by constructing the class activation map. However, it does not evaluate the significance level of the cell type–specific gene importance. Identifying gene modules autonomously and efficiently based on cell phenotype information is conducive to understanding the mechanism of cell state transitions and the regulation of different cell phenotypes [18,19].

With the development of deep learning techniques, many methods have been developed to extract low-dimensional features from high-dimensional single-cell data and integrated single-cell multi-omics data in a low-dimensional space. Cobolt constructed a multimodal variational autoencoder based on a hierarchical Bayesian generative model that projected the single-cell multi-omics data into shared latent space to perform visualization and clustering [20]. Single-cell Deep learning model for ATAC-Seq and RNA-seq Trajectory integration (scDART) is a deep learning framework that compresses scRNA-seq and scATAC-seq data into a shared space and aligns cells according to trajectories [21]. Graph-linked unified embedding (GLUE) also enables single-cell multi-omics data integration by encoding cells into the latent space [22]. Compared with the previous 2 methods, GLUE introduces the knowledge-based guidance graph via a graph autoencoder and extracts gene features to correct the alignment of cells

in latent space. However, these methods share a common limitation: They all align cells from different omics data within a latent space. While this facilitates cell clustering and annotation, its biological interpretations make extracting deep mechanisms from the data difficult. Also, Cobolt and scDART rely on shared information, leading to data loss. They only extract low-dimensional features of cell and do not process genes, ignoring potential relationships between genes. scDART aligns cells in low-dimensional space according to cell trajectories, which may not be applicable to single-cell data without differentiation trajectories. GLUE introduces predefined knowledge-based guidance graphs, such as protein interaction networks, introducing noise beyond single-cell data. Some statistical frameworks, such as multi-omics factor analysis (MOFA2) and a nonnegative matrix factorization algorithm (UINMF), are also designed to integrate single-cell multi-omics data [23,24]. MOFA2 builds on the Bayesian group factor analysis framework to infer a low-dimensional representation of the data in terms of a small number of (latent) factors that capture the global sources of variability. UINMF derives a nonnegative matrix factorization algorithm for integrating single-cell datasets containing both shared and unshared features. These methods still compress data into low-dimensional features for data integration, which still fails to explain the biological mechanisms in single-cell multi-omics data. Integrating multi-omics data at the pathway and gene module levels enables a comprehensive study of complex biological processes, highlights the interrelationship of relevant biomolecules and their functions, and can mine potential biological mechanisms that cannot be discovered by single-omics data [25,26]. Nevertheless, a gap remains in inferring active pathways and cell phenotype–associated gene modules supported by single-cell multi-omics data.

Hence, we proposed a uniform framework called scapGNN, which was a graph neural network (GNN)-based framework that inferred and reconstructed gene–cell, gene–gene, and cell–cell association relationships for transforming sparse single-cell profile data into the stable gene–cell association network. Furthermore, the scapGNN integrated single-cell multi-omics data, calculated single-cell pathway activity scores, and identified cell phenotype–associated gene modules by quantifying network information. The real and simulated single-cell datasets were used to benchmark the performance of scapGNN, demonstrating that it outperformed state-of-the-art methods in multiple single-cell data analysis tasks.

## Results

### Overview of the scapGNN framework

The scapGNN leveraged the GNN model with a multimodal autoencoder to convert the sparse unstable single-cell profiling data into a stable gene–cell association network to identify active pathways and cell phenotype–associated gene modules from single-cell multi-omics data of scRNA-seq and scATAC-seq. The random walk with restart (RWR) algorithm based on graph theory further inferred the pathway activity score matrix and identified cell phenotype–associated gene modules (Fig 1A and Materials and methods).

The GNN model included a preprocessed gene–cell matrix after the removal of low-quality cells and genes, normalization, and selection of highly variable features [27] or informative differentially expressed genes (DEGs) [28] for the scRNA-seq gene expression profile or gene activity matrix of scATAC-seq (Materials and methods). First, the encoder of the deep neural network autoencoder (DNNAE) learned the low-dimensional embeddings of cell and gene features from the gene–cell matrix. A matrix factorization-based decoder was used to infer the potential gene–cell association matrix [29] (Fig 1A and Materials and methods). The left truncated mixture Gaussian (LTMG) model [30] was used to extract the transcriptional regulation states (TRSs) from the gene–cell matrix through the kinetic relationships of the transcriptional

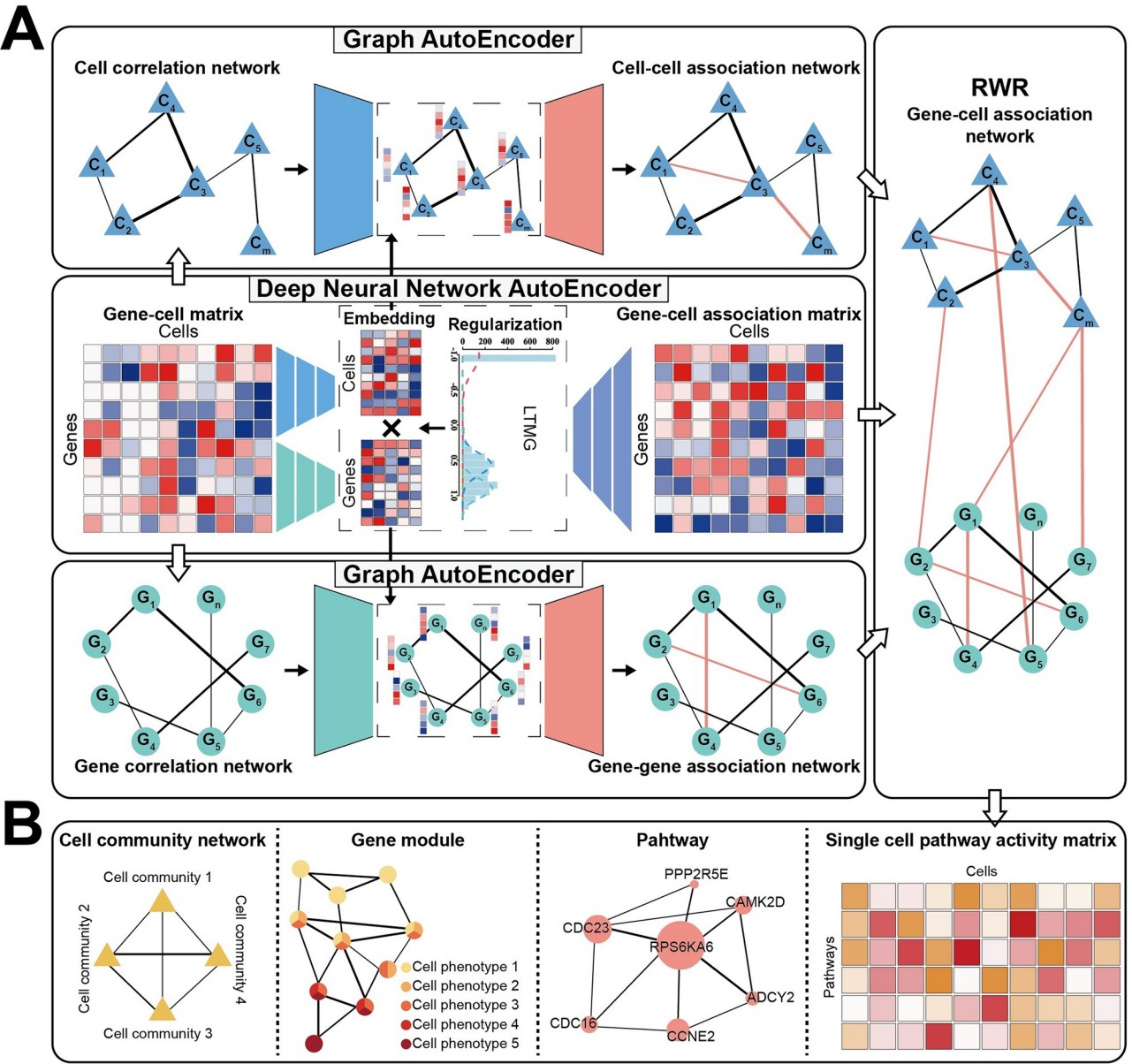

**Fig 1. An overview of the scapGNN framework.** (**A**) The input was the gene–cell matrix of scRNA-Seq or gene activity matrix generated from scATAC-seq. A graph-based autoencoder, which contained a DNNAE and a VGAE, learned the latent associations between genes and cells. The RWR algorithm quantified pathway activity and identified cell phenotype–associated gene modules. (**B**) Main capabilities of scapGNN included inferring single-cell pathway activity profiles, constructing cell cluster association networks, identifying cell phenotype–associated gene modules under multiple cell phenotypes, and quantifying the importance of genes in the pathway. DNNAE, deep neural network autoencoder; LTMG, left truncated mixture Gaussian; RWR, random walk with restart; scATAC-seq, single-cell ATAC sequencing; scRNA-seq, single-cell RNA sequencing; VGAE, variational graph autoencoder.

regulatory inputs, mRNA metabolism, and abundance in single cells (Materials and methods). The TRSs elucidated the multiple expression states of genes across single cells, and a high signal value indicated that the gene was in a true active expression state in the cell [31]. The TRS signal enhanced the signal-to-noise ratio of scapGNN, controlled the direction of neural network learning, and ensured that the inferences of gene–cell associations were based on the true state of gene expression. Therefore, we used TRSs and gene expression to construct the

loss function and learn new potential associations. In brief, the gene–cell association matrix was inferred by the DNNAE integrating the gene expression features and the multimodal kinetic relationships of the gene across single cells. A gene with high association strength in a cell indicated that it was more likely to have a high level of activity in the cell compared with other genes.

Second, we used a GNN autoencoder (GAE), which is a variational graph autoencoder (VGAE) [32] containing a 2-layer graph convolution network, to perform edge task inference in gene–gene and cell–cell correlation networks. The encoder of GAE took the embeddings of genes or cells as the features of nodes in the gene or cell correlation network that was constructed from the gene–cell matrix using the Pearson correlation coefficient similar to the Dong and colleagues' study [33] (Materials and methods). The decoder inferred new association relation to regenerate gene–gene or cell–cell association network (Fig 1A and Materials and methods). The graph autoencoder reduced the amount of spurious information encoded by reconstructing the input networks by the loss function. GAE was a framework for unsupervised learning on graph-structured data that could automatically retain high-quality node relationships and remove spurious ones. It inferred new relationships based on the shared topological features of correlation networks in aggregate and output comprehensive results that covered the entire space of information associated with the input data, while remaining agnostic to any particular view of biological function. With the GAE, scapGNN could progressively embed related genes or cells of correlation networks closer during the training process, while ensuring that unrelated genes or cells remained far apart [34].

Finally, we integrated the results of the GNN model to construct a weighted gene–cell association network (Materials and methods). The RWR algorithm used genes in the pathway as seeds to calculate adjusted probability scores that represented a proximity measure between the pathway and each cell and were used as the pathway activity scores across individual cells (Fig 1B and Materials and methods). Compared with other single-cell pathway scoring methods that only considered gene expression, scapGNN captured more biological information including gene–cell associations, cell–cell associations, and gene–gene associations when calculating pathway activity scores. For gene–cell associations, highly active pathways positioned genes closer to their corresponding cells. Cell–cell associations were also considered to further distinguish cells with highly active pathways from those with less active pathways. This allowed the pathway activity score to accurately represent the heterogeneity between cells. Gene–gene associations as a background could enhance the signal-to-noise ratio of pathway activity scores. In the same way, we set cells belonging to the same phenotype (cell type, time, and disease state) as seeds to automatically identify the cell phenotype–associated gene modules (Fig 1B and Materials and methods). The gene module is the set of genes significantly most important for the characterization of the identity of that cell phenotype. For multiple cell phenotypes, we also quantified the propensity of genes to be expressed between cell phenotypes and provided a visualization program for the network of cell phenotype–associated gene modules. We could gain insight into cell phenotype–specific genes, or genes expressed in multiple cell phenotypes, and the strength of gene–gene associations. The foundation of the scapGNN framework was to infer the gene–cell association network based on gene expression features in cells. Hence, the biological significance conveyed by the associations between genes was the coexpression regulatory relationships.

We also clustered the cells into communities using 3 community detection algorithms in the cell–cell association network (Fig 1B and Materials and methods). We merged the same cell types or identified cell communities to construct cell community networks from the cell–cell association network. The edges between nodes in this network indicated the strength of similar associations between cell types or cell communities.

We developed a network fusion method to integrate the gene–cell association networks from different types of omics data to integrate single-cell multi-omics data (S1 Fig and Materials and methods). For instance, scRNA-seq and scATAC-seq data were processed using scapGNN to generate gene–cell association networks. Two gene–cell association networks were combined using Brown's method to generate a multi-omics supporting gene–cell association network, which was processed by RWR to calculate pathway activity scores and identify cell phenotype–associated gene modules.

In summary, the method enabled scapGNN for single-cell multi-omics data to construct gene–cell association network, calculate single-cell pathway activity scores, find key active genes in the pathway, identify cell phenotype–associated gene modules and cell communities, and integrate single-cell multi-omics data (Fig 1B and S1 Fig).

## Pathway activity score of scapGNN represented cell heterogeneity

Three scRNA-seq datasets for different biological applications, including cell type, cell subtype, and time series, were used to test the performance of scapGNN in representing cell heterogeneity at the pathway level (S1 Table and S1 Text). The Uniform Manifold Approximation and Projection (UMAP) for dimension reduction [35] and t-distributed stochastic neighbor embedding (tSNE) [36] were used to visualize the cell clustering results of scapGNN and the existing state-of-the-art single-cell pathway enrichment methods (AUCell, Pagoda2, and UniPath) (S2 Table). Compared with these pathway enrichment methods, scapGNN better clustered cells within the same type more densely and separated cells of different types more distinctly for the mouse pancreas (Fig 2A and S2A Fig). For the cell subtype dataset of the human embryonic stem cell (ESC)-derived dopaminergic neurons, only scapGNN aggregated different subtypes of cells, such as ESC-derived progenitor subtype 1a (eProg1a) and 1b (eProg1b) cells into individual clusters (Fig 2B and S2B Fig). This implied that scapGNN was expected to help discover new cell subtypes. For the time series dataset of the human pluripotent stem cells (hPSCs), scapGNN could better distinguish the state of cells at different times (Fig 2C and S2C Fig). For these 3 scRNA-seq datasets, we further counted 3 cell clustering accuracy indicators [adjusted rand index (ARI), normalized mutual information (NMI), and silhouette width (SW)]. scapGNN showed higher accuracy compared with AUCell, Pagoda2, and UniPath in all of 10 state-of-the-art single-cell clustering methods (S3 Fig). We further calculated the bias-corrected mutual information (BCMI) between pseudotime inferred based on pathway activity scores and true cellular timestamps (S1 Text) [37]. As shown in S4 Fig, scapGNN could more accurately represent the true cellular temporal state compared with other methods. In addition, UniPath proposed a temporal-ordering method for single-cell pathway activity scores [14]. Therefore, we followed the UniPath method to remove cell cycle–related genes and infer temporal ordering for different single-cell pathway activity scoring methods. As shown in S5 Fig, the pathway activity scores calculated using scapGNN and UniPath performed better in characterizing the temporal ordering of cells compared with AUCell and Pagoda2. Our framework merged the same cell types or identified cell communities in the cell–cell association network to observe the strength of association between them. For the time series dataset, the association between cells at the differentiation end (36, 72, and 96 h) of the definitive endoderm cells (DECs) was stronger (S6A Fig). This indicated that the cells at these timescales had diminished differentiation drive and more similar cellular functions. Early-differentiating DEC and late-differentiating DEC tended to be in different clusters for the cell communities we identified in the cell association network (S6B Fig).

We further benchmarked scapGNN using 16 scRNA-seq datasets, including the aforementioned 3 datasets (S3 Table), 10 single-cell clustering methods (S4 Table), and 3 accuracy

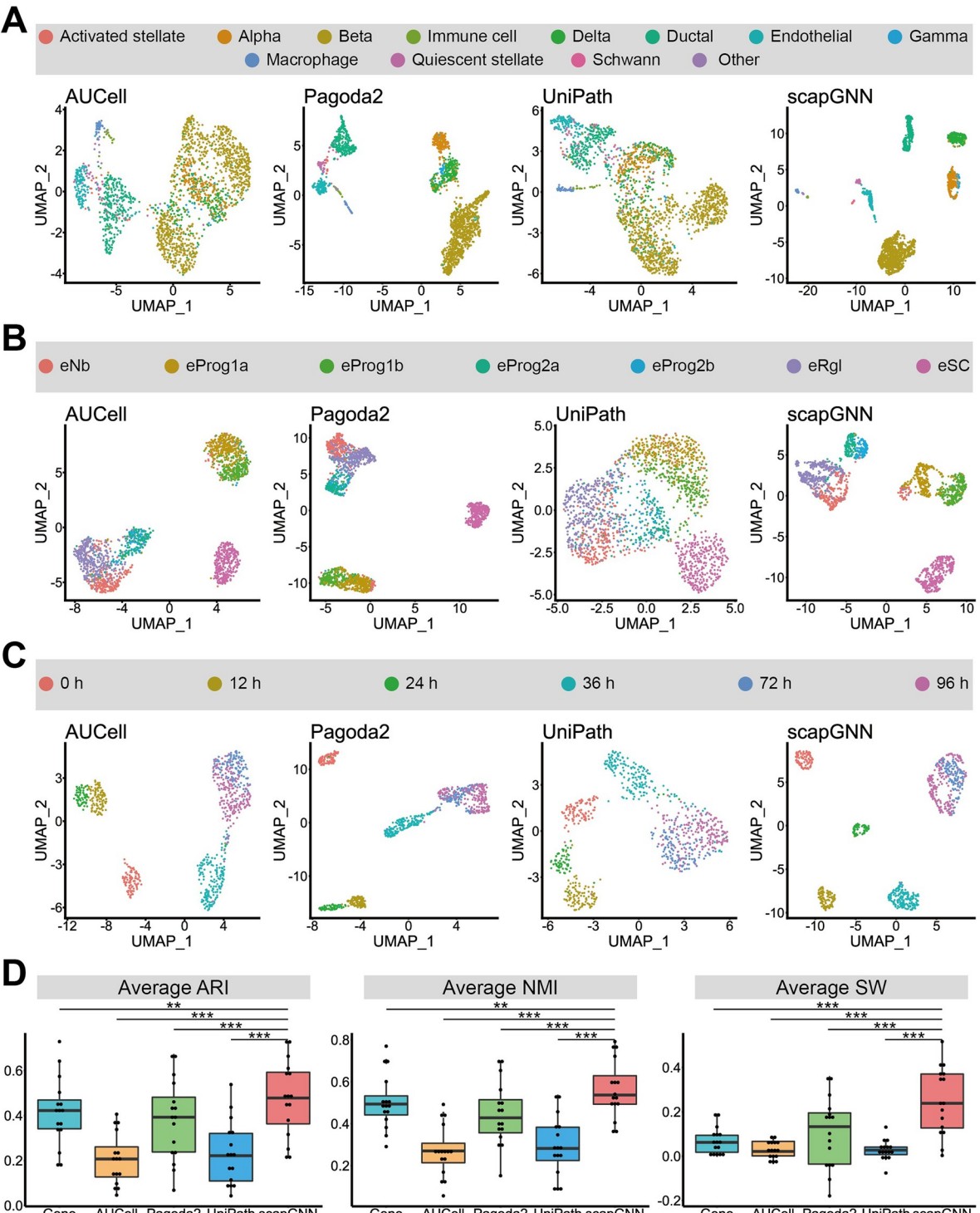

**Fig 2. Evaluation of cell clustering based on pathway activity scores.** UMAP visualizations of cell type data (**A**), time series (**B**), and cell subtype data (**C**) based on pathway activity scores using the 4 pathway enrichment methods (AUCell, Pagoda2, UniPath, and scapGNN). (**D**) Box plot of average ARI, average NMI, and average SW at the gene and pathway levels from AUCell, Pagoda2, UniPath, and scapGNN using 10 state-of-the-art single-cell clustering methods on 16 scRNA-seq data sets. The paired-sample Wilcoxon signed-rank test was used to calculate the significance $p$-values (* $p < 0.05$, ** $p < 0.01$, and *** $p < 0.001$) for the cell clustering indicator differences of each cell clustering indicator between scapGNN and other methods. The data underlying this figure can be found in S1 Data. ARI, adjusted rand index; NMI, normalized mutual information; scRNA-seq, single-cell RNA sequencing; SW, silhouette width; UMAP, Uniform Manifold Approximation and Projection.

quantification indicators to systemically evaluate the cell clustering accuracy (Materials and methods). The results revealed that scapGNN had significantly higher accuracy in cell clustering and better represented the heterogeneity of cells compared with the other methods (Fig 2D). Although gene and pathway levels are different aspects, scapGNN still had better cell clustering performance (Fig 2D). Consistent with Su's study [10], traditional bulk RNA-seq pathway enrichment analysis methods performed poorly on single-cell datasets. They were, therefore, not compared (S7 Fig). The average detected gene number in cells of each scRNA-seq dataset was determined, and the 16 benchmark datasets were categorized into 2 groups (8 high-detection gene number datasets and 8 low-detection gene number datasets) based on their median values (S8A Fig). We found that all methods in the low-detection gene number dataset had reduced cell clustering performance, but scapGNN still outperformed the other methods (S8B Fig). We also determined the values of ARI and NMI for AUCell, Pagoda2, Uni-Path, and scapGNN by 10 cell clustering methods. Overall, scapGNN performed better than other single-cell pathway activity enrichment methods (S9 Fig). The use of 16 datasets and 10 clustering methods demonstrated the scalability of scapGNN. Additionally, in scapGNN, the calculation of single-cell pathway activity scores was based on a random sampling process. The distribution of pathway activity scores was unimodal like a normal distribution, which provided a meaningful representation of cell states and applied to downstream analysis such as differential analysis or marker pathway identification [38].

We evaluated scapGNN using the gold-standard datasets with batch effects, which contained 5 cell lines and analyzed it by 3 sequencing protocols to integrate and analyze the scRNA-seq data provided by different research groups or experimental platforms [39] (S1 Table). The datasets were integrated using Seurat v4 [27] and transformed by scapGNN into a pathway activity score matrix. UMAP visualization showed that scapGNN could further reduce batch effects and improve the cell clustering accuracy relative to AUCell, Pagoda2, and UniPath (S10A and S10B Fig). Meanwhile, scapGNN had higher accuracy for identifying the correct marker gene set for A549 cells in the integrated data compared with other methods (S10C Fig).

We then tested the cell clustering performance of scapGNN on atlas-scale scRNA-seq datasets. A mouse cell atlas dataset [40] (S1 Table) was used, and the cells containing less than 800 nonzero expressed genes were filtered (set parameter min.features = 800 in Seurat). As shown in S11 Fig, scapGNN also exhibited excellent cell clustering performance in large atlas-level data.

We further assessed the uniformity of scapGNN by ablation experiments (S1 Text). For the cell type dataset, we counted cell clustering accuracy indicators of pathway activity scores from DNNAE, GAE+LTMG, and scapGNN-LTMG. In contrast, the unified scapGNN framework performed better (S12 Fig).

## scapGNN accurately identified active pathways and cell phenotype–associated gene modules

We used the cell marker gene sets of 460 cell types collected by Chawla and colleagues [14] from CellMarker [41], BioGPS [42], and Harmonizome [43] databases to evaluate the performance of pathway identification due to the lack of gold-standard pathway data in single-cell studies (Materials and methods). For homogeneous data (K562 and A549) and heterogeneous datasets (GM12878, and ESC), we counted the proportion of cells with the correct marker gene sets of the 4 cell types detected in the top 5 of the pathway activity scores (S1 Table). The results showed that scapGNN detected more substantial cells with the correct marker gene sets in both homogeneous (containing only 1 cell type) and heterogeneous (containing multiple

cell types) data compared with AUCell, Pagoda2, and UniPath (Fig 3A). Although the homogeneous data contained only 1 cell type, scapGNN could still identify the truly active gene set. This was because the gene–cell association network contained gene–cell associations in which genes with high activity were more likely to be in proximity to the corresponding cells compared with other genes. We collected the scRNA-seq datasets that contained T and B cells to

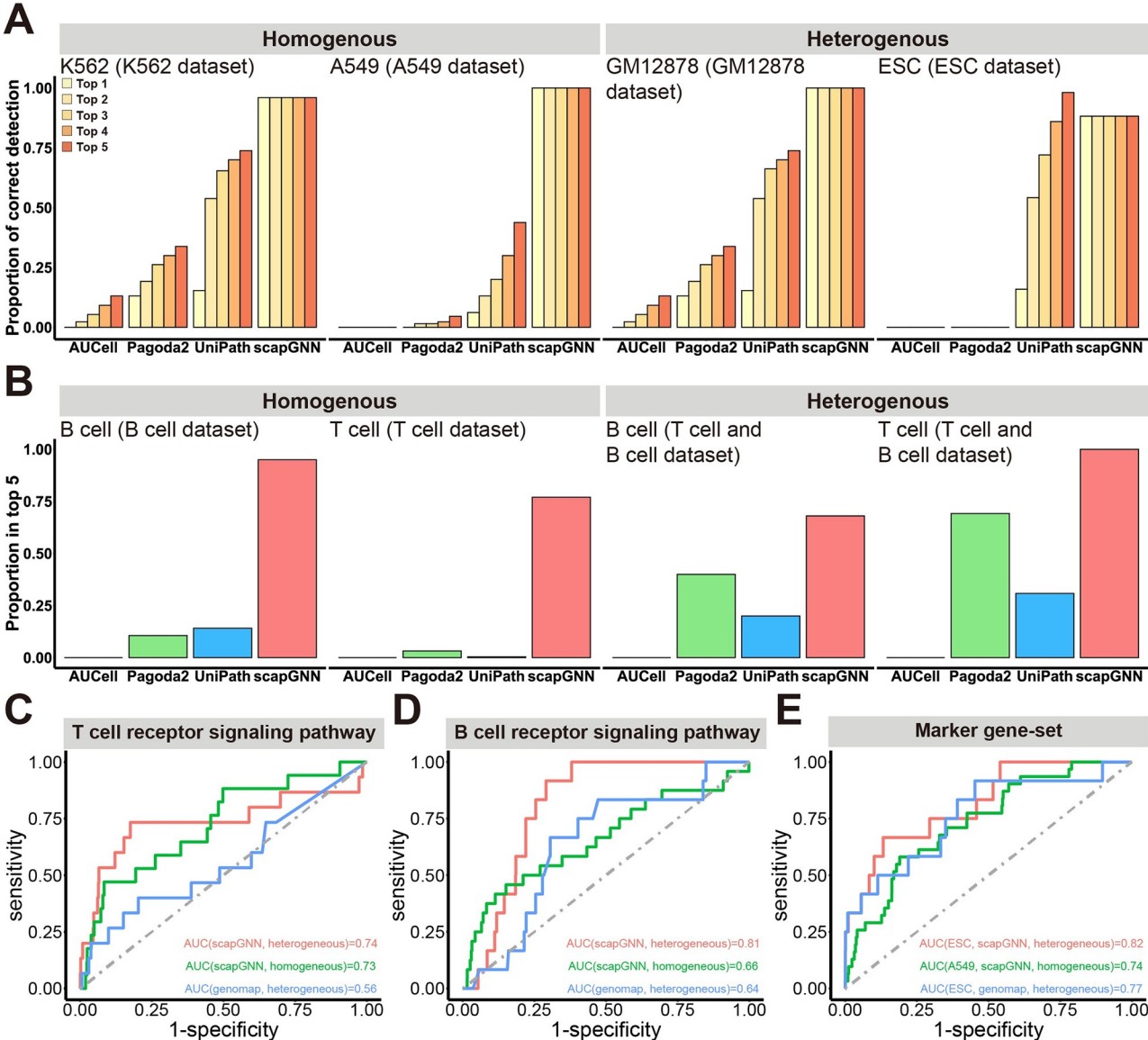

**Fig 3. Accuracy evaluation of scapGNN in identifying pathway and cell phenotype–associated gene modules.** (**A**) Proportion of cells that detected the correct marker gene sets using scapGNN and the other 3 methods in the top 1–5 of the pathway activity scores on both homogeneous and heterogeneous scRNA-seq datasets. (**B**) Accuracy of 4 pathway enrichment methods for identifying gold-standard pathways. The proportion of cells with T- or B cell receptor signaling pathway in the top 5 pathways. (**C**) ROC curves of T cell–associated gene modules in T cell (homogeneous) or T- and B cell (heterogeneous) scRNA-seq datasets using genes of the T-cell receptor signaling pathway as the gold standard. (**D**) ROC curves of B cell–associated gene modules in B (homogeneous) or T cell and B cell (heterogeneous) scRNA-seq datasets using genes of the B cell receptor signaling pathway as the gold standard. (**E**) ROC curves of ESC-associated gene modules and A549-associated gene modules in the ESC (heterogeneous) and A549 (homogeneous) datasets using marker genes of the corresponding cell type as the gold standard. The data underlying this figure can be found in S2 Data. ESC, embryonic stem cell; ROC, receiver-operating characteristic; scRNA-seq, single-cell RNA sequencing.

further verify the accuracy of the pathway enrichment (S1 Table). scapGNN could more accurately identify known T- and B cell activated pathways (T- and B cell receptor signaling pathways) than AUCell, Pagoda2, and UniPath (Fig 3B). The true active pathways (T- and B cell receptor signaling pathways) of T and B cells tended to have high-activity scores (S13A and S13B Fig). We used Seurat to identify marker pathways in T and B cells and found that the T- and B cell receptor signaling pathways could be identified (adjusted $p$-value $< 0.0001$) and ranked first in ascending order of adjusted $p$-values (S13C and S13D Fig). We assessed the stability of scapGNN pathway scoring by grouping T or B cells with different cells. The results showed that scapGNN could consistently identify the T- and B cell receptor signaling pathways, even in different combinations of datasets (S14 Fig). This suggested that although scapGNN used cell–cell association to enhance the ability of the pathway activity score to discriminate between cells, it did not affect the proximity of genes contained in the true active pathway to the corresponding cells. This also highlighted the advantage of scapGNN integrating more biological information to calculate pathway activity scores.

Next, we evaluated the accuracy of scapGNN in identifying cell phenotype–associated gene modules. For homogeneous and heterogeneous data, we calculated the association scores of cell types with genes using cells belonging to the same type as a seed. These association scores were sorted in descending order. Using the marker genes of the corresponding cell type or the genes of the T- and B cell receptor signaling pathways as the gold standard, the receiver-operating characteristic (ROC) curve analysis was used to assess the accuracy of scapGNN in identifying phenotype-associated gene module. In addition, we used genomap, an entropy-based cartography method for discovering cell- and class-specific gene sets from scRNA-seq data, as a benchmark method (S2 Table) [17]. The genomap provided activation values for each gene in the specified cell type. However, the training of the genomap model relied on the known truth cell labels of the dataset. When the labels of the cell types were all the same (i.e., homogeneous data), the genomap provided zero activity values for all genes. Therefore, we could only compare the performance of scapGNN and genomap on heterogeneous data. The results, as shown in Fig 3C–3E, indicated that scapGNN could identify cell type–associated active genes using either marker pathway or marker gene as a gold standard. The genomap is also effective in identifying cell type–associated active genes, particularly for cell marker genes. Both scapGNN and genomap had good robustness for dropout noise (S15 Fig). However, the accuracy of scapGNN was better than that of genomap (Fig 3C–3E and S15 Fig). An additional advantage of scapGNN over genomap was the ability to provide significance-level $p$-values for the association scores of each gene with cell phenotypes. We further used GO terms to evaluate the functional modularity of cell phenotype–associated gene modules (S1 Text). As shown in S16 Fig, the gene modules identified by scapGNN tended to be more functionally modular.

## Robustness analysis of scapGNN

In this section, we evaluated the robustness of scapGNN under noise from different sources using sciPath [1], a performance evaluation framework for integrating pathways with single-cell data. First, we randomly converted nonzero expression values to zero to simulate dropout noise. This noise was added to the cell subtype dataset at different strengths (5%, 10%, 15%, and 20%). We then quantified robustness using the AUC for the noise proportion–cell clustering accuracy indicator. UMAP visualizations and AUC scores showed that scapGNN effectively preserved the characteristics of cells and maintained better cell clustering accuracy compared with AUCell, Pagoda2, and UniPath (Fig 4A and 4B and S17A–S17C Fig).

Further, we added Gaussian noise to the cell subtype dataset and found that scapGNN had better robustness even under different noise types compared with the other 3 methods (Fig

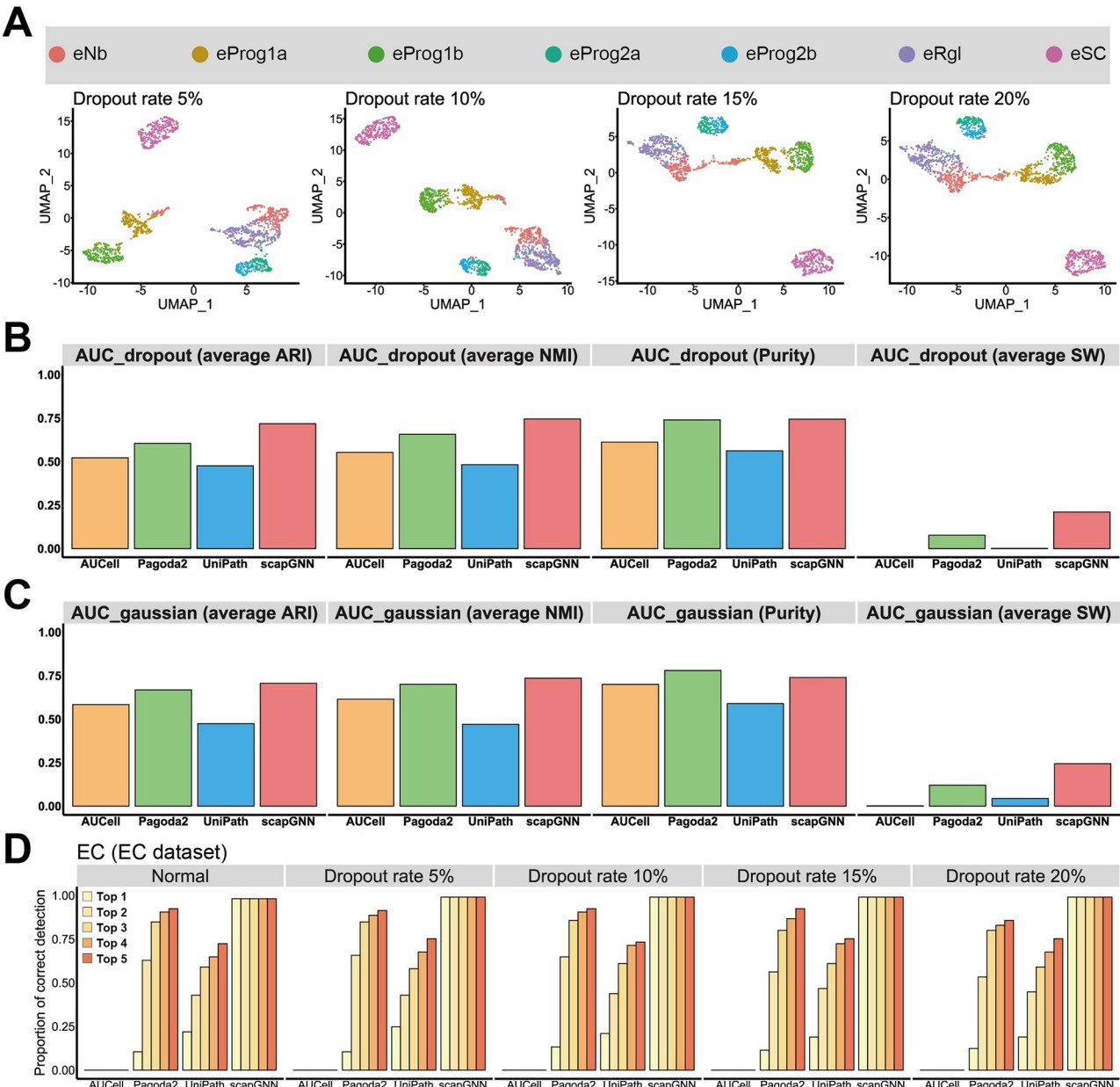

**Fig 4. Robustness evaluation.** (**A**) UMAP visualizations of the scapGNN on cell subtype dataset with different strengths of dropout noise. The AUC of 3 cell clustering accuracy indicators of AUCell, Pagoda2, UniPath, and scapGNN under dropout noise (**B**) and "Gaussian" noise (**C**). (**D**) Proportion of EC cells with corresponding marker gene set in the top 1–5 pathway scores using the 4 pathway enrichment methods under different strengths of dropout noise on the EC dataset. The data underlying this figure can be found in S3 Data. ARI, adjusted rand index; EC, endothelial cell; NMI, normalized mutual information; SW, silhouette width; UMAP, Uniform Manifold Approximation and Projection.

4C). For real data, we counted the zero-valued rate of 16 scRNA-seq datasets (S3 Table) and calculated the AUC of the zero-valued rate–cell clustering accuracy indicator curve (S17D Fig). This result indicated that scapGNN was suitable for real scRNA-seq data with different zero-value rates and showed more robust performance than AUCell, Pagoda2, and UniPath.

Next, we tested the influence of zero value and Gaussian noise on the pathway identification accuracy of scapGNN. For the endothelial cell (EC) and ESC datasets (S1 Table), we

determined the proportion of cells that detected the correct marker gene sets of corresponding cell types in the top 1 to 5 of pathway scores ranked under different noise strengths. The results indicated that the pathway identification performance of scapGNN was robust, and the accuracy was higher than that of AUCell, Pagoda2, and UniPath regarding overall noise strength (Fig 4D and S17E Fig).

## Application of scapGNN to scATAC-seq data

Besides scRNA-seq, scapGNN could also process single-cell epigenome data. We used the mouse cortical brain dataset and peripheral blood mononuclear cell (PBMC) dataset of 2 different species (mouse and human) to evaluate the performance of scapGNN in scATAC-seq data (S5 Table). scapGNN maintained high-accuracy cell clustering and pathway identification performance, which was robust to different strengths of dropout noise (Fig 5A–5C and S18A Fig). For the scATAC-seq data, scapGNN also could stably identify cell-intrinsic active pathways in combination with different cell types (S18B Fig). We next used scapGNN to identify active pathways in the scATAC-seq data of the PBMC dataset. The results of cell type–specific marker gene sets identification showed that UniPath using binomial and hypergeometric tests for pathway enrichment performed well, similar to scapGNN only on monocyte cells but failed on natural killer cells and native CD8+ T cells (Fig 5D). For the known active T-cell receptor signaling pathway in T cells, we found that scapGNN could more accurately identify the active pathways of T cells (Fig 5E). The T-cell receptor signaling pathway had a higher pathway activity score and could be successfully identified using Seurat as a marker pathway for T cells (S19A and S19B Fig). We also evaluated the stability of scapGNN pathway scoring in scATAC-seq data. scapGNN could still consistently identify the T-cell receptor signaling pathway in the top 5 pathways of T cells (S20 Fig). We tested the robustness of scapGNN in the scATAC-seq data by adding different strengths of dropout noise to the PBMC dataset. The results showed that scapGNN and the hypergeometric test method of UniPath were highly robust to the dropout noise of the scATAC-seq data (Fig 5F). Thus, scapGNN performed well in the analysis of pathway activities for scATAC-seq datasets.

## Inferring pathway activity scores by integrating scRNA-seq and scATAC-seq

Next, we tested the performance of scapGNN to integrate single-cell multi-omics data. We applied scapGNN to 3 single-cell multi-omics datasets with different sequencing platforms, tissues, and species to evaluate the ability of the single-cell multi-omics supported pathway activity scores inferred by scapGNN to represent the cellular heterogeneity (S5 Table). UMAP plots of multi-omics supported pathway activity scores showed that scapGNN could clearly distinguish between different cell types after integrating single-cell multi-omics information (S21A–S21C Fig). Compared with 5 state-of-the-art single-cell multi-omics integration methods (UINMF, MOFA2, Seurat, Cobolt, and GLUE), scapGNN showed better performance on several cell clustering indices (Fig 6A and 6B, and S21D Fig and S2 Table).

In hair follicle tissue, transit-amplifying cells (TACs) can proliferate rapidly and produce 3 different types of mature cells, including the inner root sheath (IRS), hair shaft cuticle/cortex, and medulla [44]. Recently, Jones and colleagues applied SHARE-seq, an approach that enabled the joint measurement of chromatin accessibility and gene expression from the same single cells to adult mouse skin tissue and resolved the differentiation process of TACs [44]. We applied scapGNN to integrate scRNA-seq and scATAC-seq data from the mouse skin dataset and infer the pseudotime of TAC differentiation using Monocle 3 [45]. Compared with scDART (S2 Table), scapGNN was a method for integrating scRNA-seq and scATAC-seq data

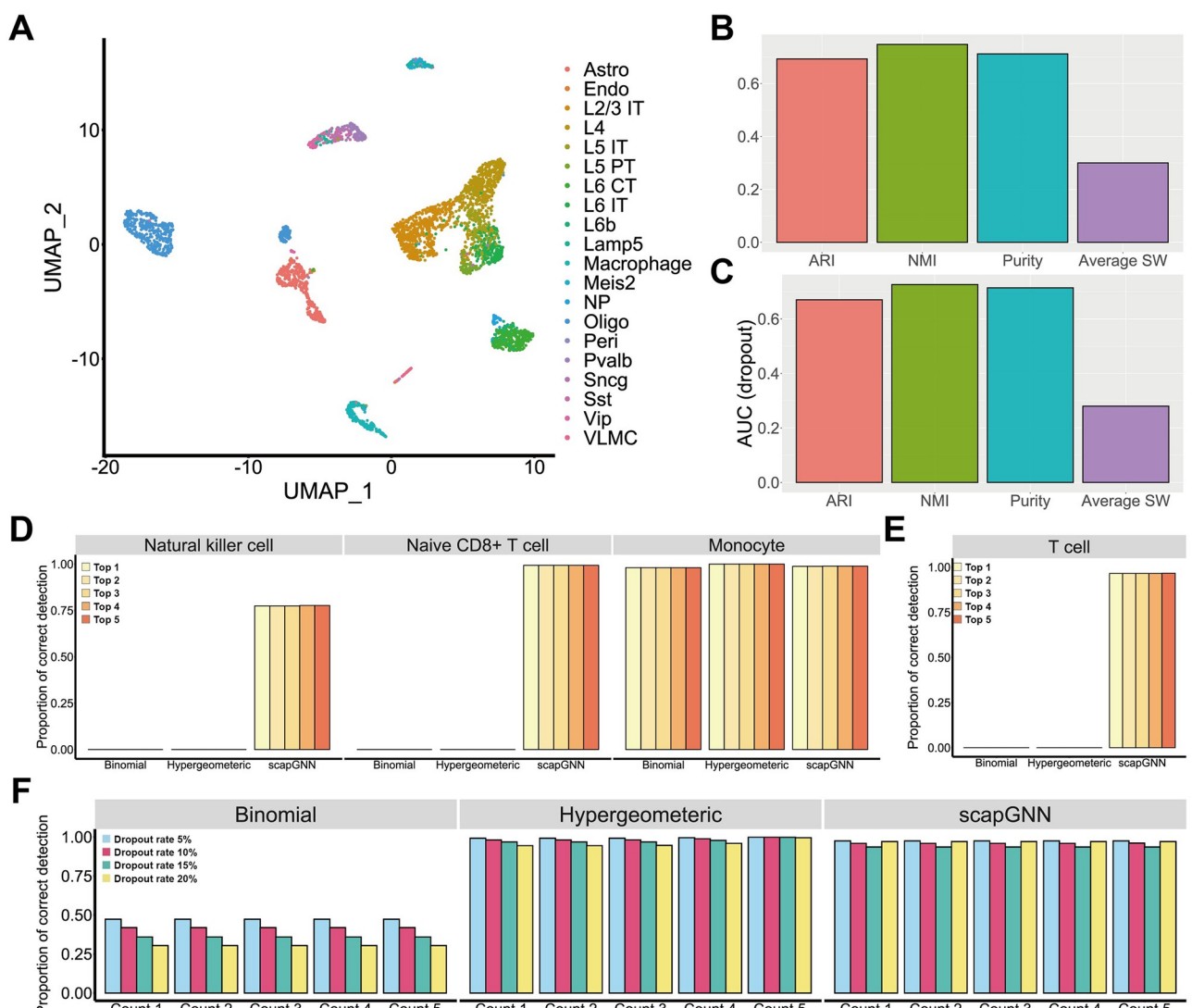

**Fig 5. Performance of scapGNN on scATAC-seq data.** (**A**) UMAP visualization of mouse cortical brain dataset using pathway activity score matrix of scapGNN. (**B**) Bar graph of the 4 cell clustering accuracy indicators for pathway activity score matrix of scapGNN on mouse cortical brain dataset. (**C**) AUC of 4 cell clustering accuracy indicators for pathway activity score matrix of scapGNN on mouse cortical brain dataset with dropout noise of different strengths. The proportion of cells that detected the corresponding correct cell type marker gene sets (**D**) and the proportion of T cells that detected the T-cell receptor signaling pathway (**E**) in the top 1 to 5 of the pathway scores on the PBMC dataset. (**F**) Robustness evaluation of the scapGNN in correctly detecting the marker gene set of monocytes with different dropout rates on the PBMC dataset. The data underlying this figure can be found in S4 Data. ARI, adjusted rand index; AUC, area under the recovery curve; NMI, normalized mutual information; PBMC, peripheral blood mononuclear cell; scATAC-seq, single-cell ATAC sequencing; SW, silhouette width; UMAP, Uniform Manifold Approximation and Projection.

and could be used for trajectory inference on integrated data. It showed the differentiation trajectory of TACs more accurately (Fig 6C and S22 Fig). The pseudotime based on single-cell multi-omics–supported pathway activity scores calculated using scapGNN had a higher similarity to the TAC differentiation pseudotime provided by Jones's study [44] compared with scDART (Fig 6D).

We discovered that some significantly different pathways essential for key biological functions of astrocytes and oligodendrocytes in the mouse brain cortex dataset were identified only through multi-omics data integration (S6 Table). For T and B cells in the PBMC multi-omics

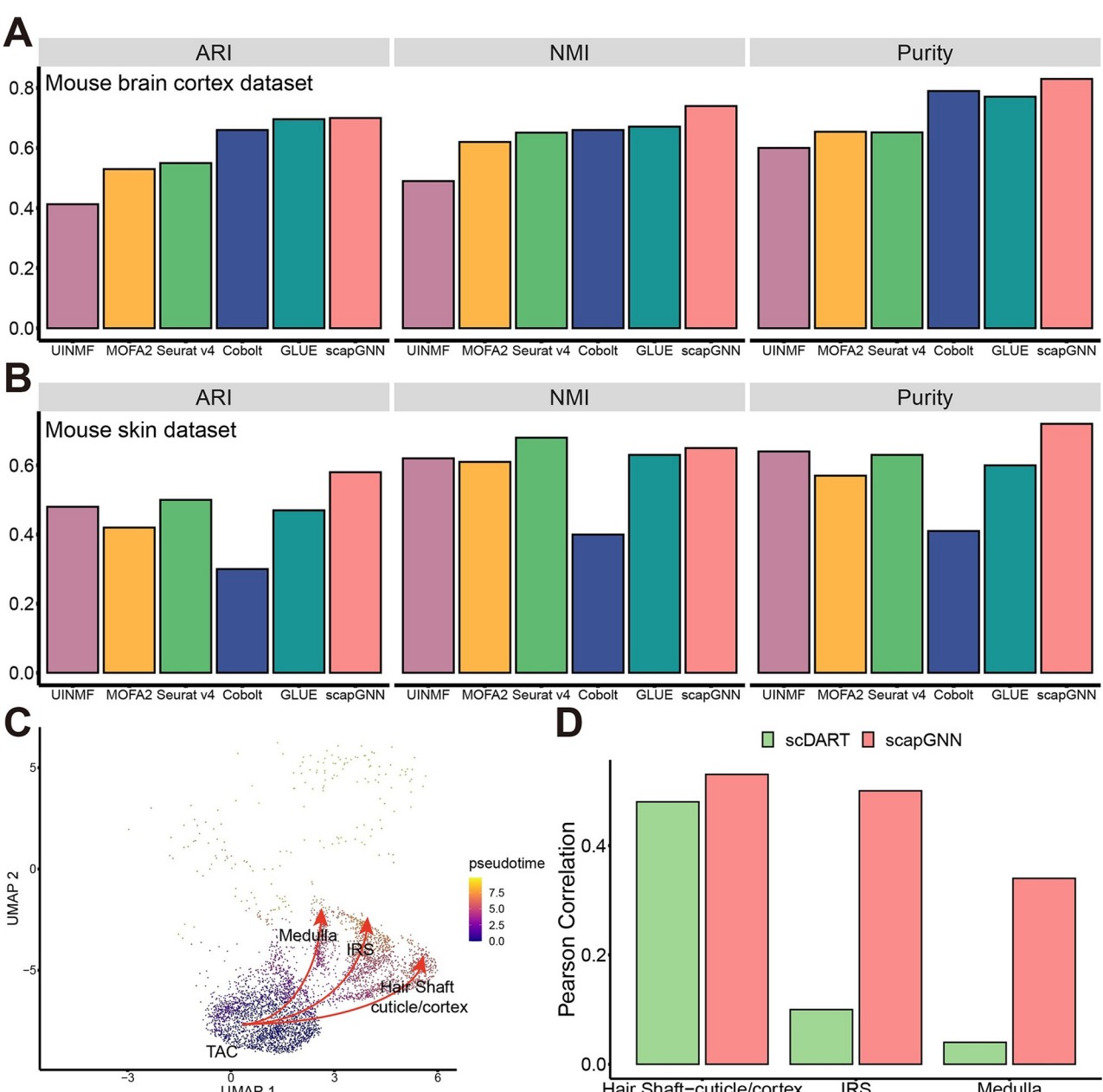

**Fig 6. Performance of scapGNN on single-cell multi-omics data integration.** Bar graph of the 3 cell clustering accuracy indicators for scapGNN, and state-of-the-art single-cell multi-omics integration methods on the mouse brain cortex dataset (**A**) and mouse skin dataset (**B**) (the resolution parameter of Seurat was 0.5). (**C**) UMAP plot colored by the pseudotime of TACs, medulla, IRS, and hair shaft cuticle/cortex cells. Red arrows are TAC populations pointing to the medulla, IRS, and hair shaft cuticle/cortex cells. (**D**) Bar graph of the Pearson correlation coefficient between the pseudotime of scapGNN or scDART and Jason's study. The data underlying this figure can be found in S5 Data. ARI, adjusted rand index; GLUE, graph-linked unified embedding; IRS, inner root sheath; NMI, normalized mutual information; scDART, single-cell Deep learning model for ATAC-Seq and RNA-seq Trajectory integration; TAC, transit-amplifying cell; UMAP, Uniform Manifold Approximation and Projection.

dataset, the proportion of cells with T- or B cell receptor signaling pathway in the top 5 of the single-cell multi-omics–supported pathway activity score list was more than 0.95 (S23A Fig). For single-cell multi-omics data integration, scapGNN could maintain the stability of pathway scoring (S24 Fig). Using genes in the T- or B cell receptor signaling pathway as the gold

standard, the gene modules identified using scapGNN for T or B cells could accurately identify these genes with AUCs above 0.8 (S23B Fig). A consistent trend was observed in the results for marker gene sets, with scapGNN being able to accurately identify the marker gene set corresponding to the cell type in both heterogeneous and homogeneous data (S23C Fig). ROC curves showed that scapGNN could accurately identify highly active genes in the gene module using the marker gene set as the gold standard (S23D Fig). A number of studies showed that the hedgehog signaling pathway was required for TAC developmental processes [44,46,47]. In particular, the transcriptome and epigenome activities of Gli2 and Gli3 in the hedgehog signaling pathway played a critical role [46,47]. This was confirmed by the violin plots of Gli2 and Gli3 activities in RNA and DNA in the mouse brain cortex dataset (S25A and S25B Fig). Consistent with this, scapGNN also detected high single-cell multi-omics–supported pathway activity scores of the hedgehog signaling pathway during TAC development (S25C Fig). Gli2 and Gli3 were coexpressed during TAC development in the gene modules identified based on the combined gene–cell association network (S25D Fig).

These results demonstrated that scapGNN effectively integrated single-cell multi-omics information at the pathway level to cluster cells, infer cell differentiation processes, and accurately identify highly active pathways and cell phenotype–associated gene modules.

## Applications of scapGNN in spermatogenesis and early embryo development

Development-related biological events involve cell differentiation and intercellular regulation [48,49]. Unraveling the biological mechanisms involved is still challenging. Therefore, we applied scapGNN to the scRNA-seq datasets of mouse spermatogenesis, early embryo development, and human testis (S1 Table).

First, Monocle 3 [45] was used to construct single-cell trajectories based on the pathway activity score matrix. The results showed that scapGNN could clearly distinguish different cell types and better reconstruct the process of spermatogenesis and early embryonic development (Fig 7A and S26 Fig). We then identified the differentially active pathways between the cell types in the mouse spermatogenesis dataset based on Seurat and filtered the pathways by p. adj < 0.001 (S6 Data). We found that the oxidative phosphorylation pathway had higher activity from early pachytene (eP) spermatocytes to early round spermatids (RS2 to RS4) (Fig 7B). After differentiation of spermatogonia into spermatocytes, meiotic spermatocytes crossed the blood–testis barrier and became dependent on lactate secreted by Sertoli cells for energy production. Lactate was oxidized to pyruvate and transported into mitochondria to fuel the oxidative phosphorylation pathway [50]. We also found that cytochrome-c oxidase-related genes of the oxidative phosphorylation pathway were of higher importance (S27A Fig), and they were shown to be highly expressed in spermatocytes and spermatids [51,52]. In addition, we identified the pathways that varied over the trajectory of the early embryo development based on a Monocle-fitted generalized linear model and q-values < 0.001 (S6 Data). We found that the peroxisome proliferators–activated receptor (PPAR) signaling pathway was significantly dynamically altered in early embryo development in mice (Fig 7C). Defects in the PPAR signaling pathway led to significantly delayed early embryo development [53]. The observed trends of the PPAR signaling pathway were consistent with the data of EmExplorer, which is an experimentally supported database for mammalian embryos [54] (S27B Fig).

Next, we characterized the developmental stage–associated gene modules. Several genes in our identified RS8-specific gene modules, such as protamine 1 (*Prm1*), protamine 2 (*Prm2*), the outer dense fiber of sperm tails 2 (*Odf2*), histone linker H1 domain (*Hils1*), and transition protein 1 (*Tnp1*), have been previously reported to be specifically expressed in round

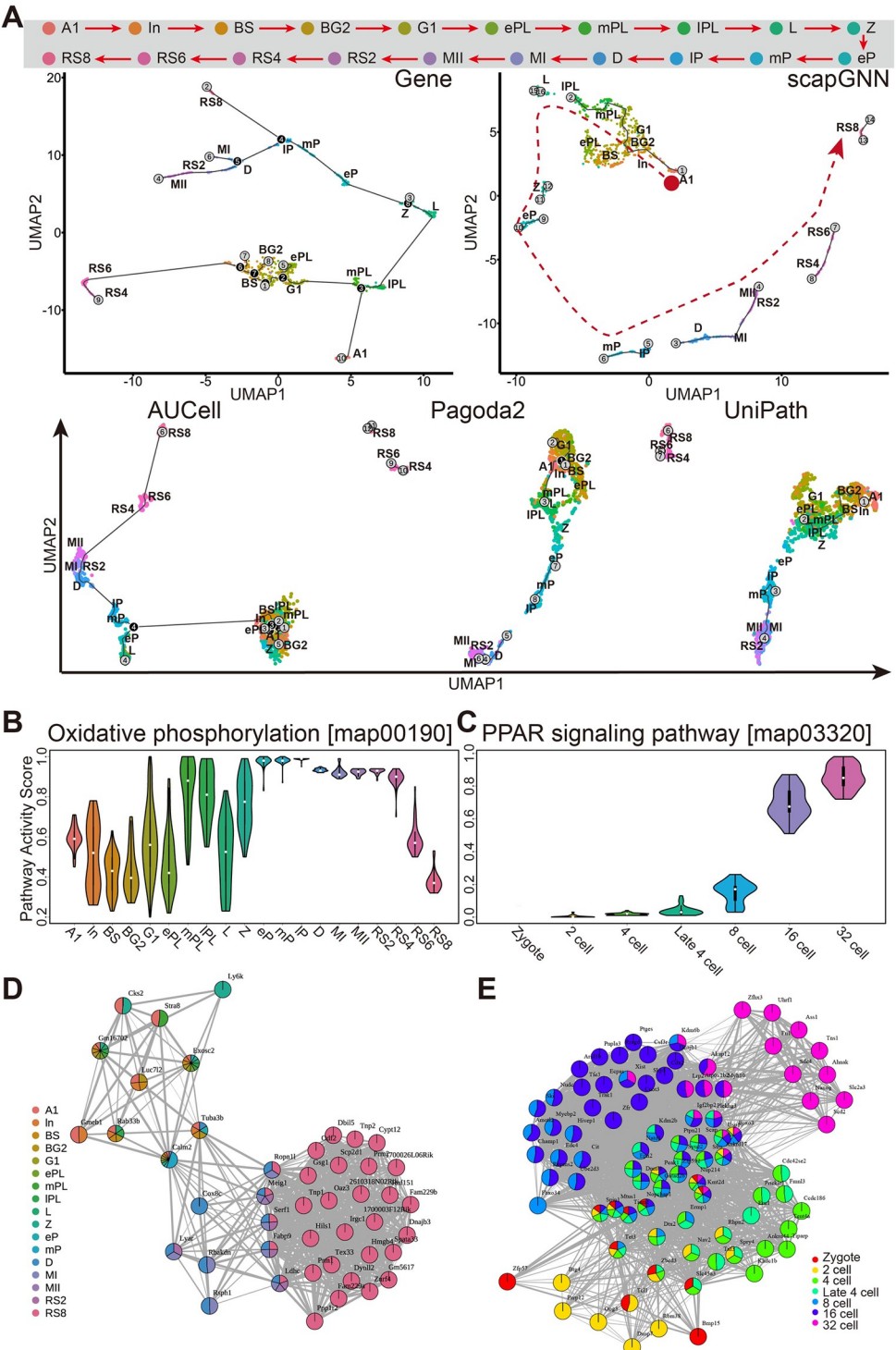

**Fig 7. Analysis of cell differentiation trajectory, dynamic pathways, and gene modules in mouse spermatogenesis and early embryonic development using scapGNN.** (**A**) Cell differentiation trajectory analysis of 20 spermatogenesis stages using the gene expression data, and scapGNN, AUCell, Pagoda2, or UniPath-based pathway activity score matrix on mouse spermatogenesis dataset via Monocle 3. At the top, the solid red arrows indicate the order of the 20 stages of spermatogenesis. The 20 stages of spermatogenic cells include A1, type A1 spermatogonia; In, intermediate spermatogonia; BS, S-phase type B spermatogonia; BG2, G2/M-phase type B spermatogonia; G1, G1-phase preleptotene; ePL, early S-phase preleptotene; mPL, middle S-phase preleptotene; lPL, late S-phase preleptotene; L, leptotene; Z, zygotene; eP, early pachytene; mP, middle pachytene; lP, late pachytene; D, diplotene; MI, metaphase I;

MII, metaphase II; RS2, steps 1–2 spermatids; RS4, steps 3–4 spermatids; RS6, steps 5–6 spermatids; and RS8, steps 7–8 spermatids. In the scapGNN panel, red arrows indicate the order of differentiation of the spermatogenic cells. (**B**) Box plots of activity scores of the oxidative phosphorylation pathway, which was significantly expressed in the early round spermatids (RS2 to RS4) of spermatogenesis and (**C**) the PPAR signaling pathway, which was significantly dynamic in the developing mouse embryos. Network of cell phenotype–associated gene modules during mouse spermatogenesis (**D**) and embryo development (**E**). In the network, the sector area of a node indicates the strength of association between a gene and a cell phenotype, and the width of the edges between nodes indicates the strength of association between genes. The data underlying this figure can be found in S6 Data.

spermatids [55–57] (Fig 7D). For early embryo development, B cell translocation gene-4 (*Btg4*) was essential and its deletion led to 1- or 2-cell arrest in the gene module specifically associated with the 2-cell embryos [58] (Fig 7E). Additionally, we found that some genes associated with multiple cell types were involved in the transition of germ cells and the maintenance of normal spermatogenesis. For example, *Fabp9* was highly coexpressed from the metaphase I (MI) spermatocytes to the round spermatid stage and was essential for the formation of the normal shape of the sperm head [59]. Among the network of spermatogenesis stage–associated gene modules, RBAK downstream neighbor (*Rbakdn*) and spermatogenesis-associated 33 (*Spata33*) were the 2 genes with the highest association scores with *Fabp9*, and their expression patterns showed a clear co-occurrence with *Fabp9* (S28A and S28B Fig). The deletion of Spata33 was also reported to cause abnormalities in sperm formation with sperm midpiece defects and infertility [60]. In comparison, cysteine-rich perinuclear theca 12 (*Cypt12*) and high-mobility group box 4 (*Hmgb4*) were 2 genes with a lower strength of association with *Fabp9*; they did not have a consistent expression trend with *Fabp9* (S28C and S28D Fig).

Finally, we constructed the cell community network for Sertoli cells and spermatogenic cells from the human testis dataset (S27C Fig and Materials and methods). We found that the Sertoli cells were tightly connected to spermatogenic cells. The cell phenotype–associated gene networks discovered the regulatory mechanisms between cell types. Clusterin (*CLU*) and pro-saposin (*PSAP*) are produced by Sertoli cells [61,62] and are the connecting nodes of the gene modules of Sertoli cells and spermatogenic cells (S27D Fig). *CLU* regulates the meiosis of germ cells, protects the testes from heat stress–induced injury, and ensures the normal progress of spermatogenesis [63,64]. *PSAP* acts as glycolipid transfer between Sertoli cells and the developing spermatids, and its abnormalities can lead to delayed timing in sperm cell development [62,65,66].

## Investigation of the biological mechanisms in coronavirus disease 2019 based on scapGNN

Next, the scapGNN was applied to the coronavirus disease 2019 (COVID-19) dataset, which included healthy controls and patients with COVID-19 (S29A Fig and S1 Table). We screened B cell–associated differential pathways between healthy controls and patients with COVID-19 by taking the top 10 fold-change for each B cell subtype in descending order, with the maximum $p$-value $< 5 \times 10^{-5}$ (S30 Fig). Coronavirus, as an exogenous virus, activated the host immune system. As expected, many differential pathways were associated with infections and immune diseases, such as human cytomegalovirus infection and measles, suggesting that B cells were activated to produce an immune response to exogenous viruses. Consistent with previous studies, protein complex assembly and protein transport–related pathways were activated in the B cells of patients with COVID-19, as B cells require high levels of protein synthesis to perform their functions [67]. The activity of thyroid hormone, a substance upstream of plasma cell

activation, and the activity of the thyroid hormone synthesis pathway significantly increase in patients with COVID-19 (S30 Fig). Several reports showed that thyroid hormone had a protective effect in patients with COVID-19, and low serum levels might increase mortality [68,69]. Notably, the gene association network of the thyroid hormone synthesis pathway showed that *ATP1B3* played a key role in the pathway activity in patients with COVID-19 (S29B Fig). Liu's study also found *ATP1B3* to be one of the targets of COVID-19 infection [70].

We further integrated the scRNA-seq data of influenza A virus (IAV)-infected patients and constructed networks of gene modules for B cells of 3 cell phenotypes (healthy control, COVID-19, and IAV) using false discovery rate (FDR) < 0.05 (S29C Fig). Genes coexpressed in the 3 cellular phenotypes might be disease independent. COVID-19-specific genes might be potentially high-activity gene signatures of COVID-19. For example, *NPM1* was present in the COVID-19-associated protein–protein interaction (PPI) network module and associated with enhanced viral replication of COVID-19 [71,72]. COVID-19 and IAV coexpressed genes might be associated with essential biological functions of B cells; for example, *IGHG3* and *IGLC2* were B cell marker genes [73]. Interestingly, the absence of *ARPC1B*, a health-specific gene reduced in COVID-19, led to immunodeficiency [74,75]. This might provide new insights into the pathogenic mechanisms and treatment of COVID-19.

## Discussion

Pathway enrichment analysis is a wide framework for condensing information from gene expression profiles into a pathway or signature summary [8]. Single-cell data have a large number of dropout events, leading to difficulties in data analysis at the gene level [76]. Most single-cell clustering methods only use genes as features of cells, ignoring the relationship between genes, which could make clustering methods more susceptible to noise, resulting in low accuracy and robustness. Pathway-level analysis is less affected by the noise on a single gene [1] and can provide better biological interpretation and insights [6]. In this study, we introduced scapGNN, a GNN-based framework, to transform single-cell data into the pathway activity score matrix and identified the cell phenotype–associated gene modules.

scapGNN effectively addressed the 4 major questions of pathway enrichment in single cells. The first question was whether the pathway activity score could characterize single-cell heterogeneity and accurately identify cell clusters. We benchmarked the performance of scapGNN using 16 scRNA-seq datasets, 10 state-of-the-art single-cell clustering methods, and 3 cell clustering accuracy indicators (ARI, NMI, and SW). scapGNN could capture subtle differences between cells and more accurately cluster cells in low-dimensional space compared with AUCell [12], Pagoda2 [13], and UniPath [14] (Fig 2). scapGNN also performed better in handling scRAN-seq data with batch effects (S10 Fig). The second question was whether the pathway activity scores represented the true pathway states. We applied scapGNN to homogeneous and heterogeneous datasets to detect whether cell marker gene sets and known activated pathways of T and B cells had high scores in the corresponding cells. The result indicated that scapGNN had high precision in pathway identification ability compared with the existing single-cell pathway enrichment methods (Fig 3A and 3B). The third question was whether scapGNN was robust. We simulated the noise with different strengths and types. scapGNN maintained accurate cell clustering and pathway identification capabilities (Fig 4 and S17 Fig). The final question was integrating single-cell multi-omics information into the pathway level. We proposed a network fusion method to infer pathway activity score matrix supported by multi-omics information. In terms of single-cell multi-omics data integration, scapGNN outperformed the state-of-the-art single-cell multi-omics integration methods in terms of cell clustering and cellular temporal inference and still could accurately identify pathways (Fig 6,

S22 and S23 Figs). After integrating single-cell multi-omics data, scapGNN could find pathways with differential activity that could not be identified based on single-omics alone (S6 Table). Finally, scapGNN also could identify cell phenotype–associated gene modules, which filled the gap in gene module analysis methods for single-cell data.

We further applied scapGNN to real biological questions, which analyzed the pathways and gene modules associated with spermatogenesis and early embryo development. Pseudotime analysis showed that scapGNN could better represent the real development process (Fig 7A and S26 Fig). Using scapGNN to analyze pathways and gene modules in numerous studies, we found that dynamically changing pathways were crucial for normal spermatogenesis. Additionally, activated gene modules revealed close regulation between Sertoli and spermatogenic cells. The scapGNN was also applied to the COVID-19 dataset, which included healthy controls and patients with COVID-19. We elucidated biological pathways closely associated with the onset and development of COVID-19 and identified specific features of COVID-19 compared with IAV as well as potential therapeutic targets (S29 and S30 Figs).

However, scapGNN still had some limitations. First, we examined the running time of the GNN module of scapGNN and the calculation time of the pathway activity scores for AUCell, Pagoda2, UniPath, and scapGNN in scRNA-seq datasets at different scales (S1 Text). The results showed that scapGNN's runtime was higher than that of the other methods, and UniPath's runtime was the shortest (S31 Fig). We believed that 3 main factors accounted for the time consumption of scapGNN: (1) scapGNN was an end-to-end framework that required the inference and learning of gene–cell association networks for each input data; (2) gene–cell association network contained more information than cells by gene expression matrix; and (3) we corrected the pathway activity scores to remove the effect of noise by perturbation analysis, which randomly selected gene nodes as seeds for the RWR. Therefore, for each data, scapGNN needed to have a basic runtime, after which an increase in cells brought about a lesser increase in runtime. However, we performed graphics processing unit computing and multi-core parallelism in scapGNN programs, providing the user with options to effectively accelerate the runtime. scapGNN could only integrate multi-omics data from the same cells. Nevertheless, multi-omics information within the same cells could fully recapitulate the true biological state [77,78]. Single-cell sequencing technologies capable of detecting multiple omics in the same cell are still an emerging field with development of platforms such as SNARE-seq [77] and CITE-seq [79]. These technologies provided the basis for the application of scapGNN (Fig 6A and 6B, S21D and S21E Fig). It should be noted that based on the 3 cell clustering accuracy indicators (ARI, NMI, and SW), the quality of scATAC-seq data is inferior compared with that of the corresponding scRNA-seq data from multi-omics of the same cell, posing a challenge for methods that integrate single-cell multi-omics data (S21E Fig). Theoretically, the number of omics data types integrated by scapGNN has no limit. Therefore, scapGNN is expected to process scNOMeRe-seq [80] and NEAT-seq [81], which enables the simultaneous profiling of more omics information in the same individual cell. Finally, scapGNN can convert gene expression profile data into a gene–cell association network, stably representing gene–cell relationships. Although the gene–cell association network is a kind of correlation network that cannot describe the communication and causality between cells, it provides a new view that uses graph theory-based methods to analyze single-cell profiling data.

## Materials and methods

### scapGNN availability

The scapGNN has been implemented as an R package and is freely available from the comprehensive R archive network (CRAN) (https://github.com/XuejiangGuo/scapGNN), GitHub

(https://cran.r-project.org/web/packages/scapGNN/index.html), and FigShare (https://figshare.com/articles/software/scapGNN/23734017). The R packages for scapGNN and related scripts are also available from Zenodo (https://doi.org/10.5281/zenodo.8322402).

## Data preprocessing

scapGNN can take single-cell epigenome or transcriptome data as input. The data need quality control and normalization to ensure their quality and usability. For scRNA-seq datasets, the cells containing more than 1% of genes with nonzero expression and those with nonzero expression in more than 1% of cells are preserved. The global-scaling normalization method (LogNormalize) was used to normalize the gene expression measurements and log-transform the result [27].

To assess the impact of highly variable gene selection on the performance of scapGNN, we performed benchmark experiments on different numbers of highly variable genes. For heterogeneous single-cell data, cell clustering accuracy and marker gene set scoring decrease as the number of genes increases (S32 Fig). Because low amounts of mRNA in individual cells, inefficient mRNA capture, as well as the stochasticity of mRNA expression can lead to dropout events in single-cell data [82], and not all genes contribute to cell-to-cell differences [83], too many genes can introduce technical noise. We selected 2,000 highly variable genes in processing heterogeneous single-cell data. For homogeneous data, there is little variation in expression levels between cells for inherently expressed genes such as marker genes. A small number of highly variable genes can cause a loss of information (S33 Fig). Our results show that 8,000 highly variable genes can represent information about intrinsic biological pathways or marker gene sets in cells while minimizing the introduction of noise (S33 Fig). We selected 8,000 highly variable genes in processing homogeneous single-cell data.

For scATAC-seq datasets, we estimated the gene activity by measuring ATAC-seq counts in the 2-kb upstream regions and gene body [84]. Subsequently, filtering, quality control, and normalization were the same as those for scRNA-seq. For other single-cell omics data, we needed to associate omics with genes and convert them into gene–cell matrices based on the corresponding evidence [25,85]. This study used 3,000 highly variable genes for scATAC-seq data and single-cell multi-omics integration studies.

## Deep neural network autoencoder

We first used the LTMG [30] model to parse the regulatory signals from gene expression. The LTMG model set a latent experimental resolution threshold $Z_{cut}$ to divide the gene expression of $N$ cells into 2 parts. The left truncated gene expression $X = \{x_1, \ldots, x_M\}$, where $X < Z_{cut}$ had zero- or low-expression values. The other part was active gene expression $X = \{x_M, \ldots, x_N\}$, where $X \geq Z_{cut}$. The probability density function of the normalized gene expression values was further modeled as a mixture of Gaussian distribution with $K$ Gaussian distributions, corresponding to $K$ transcriptional regulatory states:

$$P(X|\Theta) = \prod_{j=1}^{M} \sum_{i=1}^{K} a_i P_i(x_j|\theta_i) \times \prod_{j=M+1}^{N} \sum_{i=1}^{K} a_i \frac{1}{\sqrt{2\pi}\sigma_i} e^{\frac{-(x_j-\mu_i)^2}{2\sigma_i^2}} \qquad (1)$$

where $\Theta$ denotes the $K$ Gaussian distributions, and $a_i$, $\mu_i$, and $\sigma_i$ are the mixing probability weight, mean, and standard deviation, respectively. The expectation-maximization algorithm can estimate $\Theta$ and calculate $Z_{cut}$. The number of Gaussian distributions $K$ is defined by the Bayesian information criterion. Ultimately, discrete TRS signal values $\{0, 1, 2, \ldots, K\}$ are generated for each gene over cells. The signal value $K$ indicates that the gene expression value

belongs to the $K$ Gaussian peaks, corresponding to $K$ expression states. A high $K$ value indicates that the gene is more likely to be in a truly active expression state in the cell.

Next, we constructed a DNNAE regularized by TRSs to learn the latent associations between genes and cells. Taking a gene–cell matrix $X$ with $m$ genes and $n$ cells as input, the neural network encoder performed column compression and row compression to generate low-dimensional representations of genes and cells:

$$E_{m \times d}^g = \sigma\left(W_g^l X\right) \tag{2}$$

$$E_{n \times d}^c = \sigma\left(W_c^l X^T\right) \tag{3}$$

where $W_g^l$ and $W_c^l$ represent the learnable weight of the $l$th hidden layer for gene embedding and cell embedding. $E_{m \times d}^g$ and $E_{n \times d}^c$ denote the encoded $d$ dimensional feature matrix of genes and cells, respectively, and $\sigma$ is the nonlinear activation function.

We further used a matrix factorization decoder to obtain the potential gene–cell association matrix:

$$X' = E_{m \times d}^g \times \left(E_{n \times d}^c\right)^T \tag{4}$$

and took a mean square error regularized by a transcriptional regulatory signal as the loss function. Regularization aimed to improve the signal-to-noise ratio by adding different constraints to each gene during the learning process. The loss function was as follows:

$$loss = (1 - a)||X' - X||_2^2 + a\left(\sum (X' - X)^{2 \circ} S_{TRS}\right) \tag{5}$$

where $a$ is the regularization weight and $a \in [0,1]$. $\circ$ denotes element-wise multiplication. $S_{TRS} \in R^{m \times n}$ is the transcriptional regulatory signal matrix. According to our benchmark experiment, the learning rate was set as 0.001 and the number of iterations was 1000 (S34A and S34B Fig).

## Graph autoencoder

We first calculated the gene–gene Pearson correlation matrix $P_{m \times m}^g$ from the gene–cell matrix $X_{m \times n}$. For gene $i$, we designed the empirical $p$-values to evaluate the relative strength of the correlation:

$$p\text{-}values = \frac{|P_i^g > P_{ij}^g|}{|P_i^g > 0|}, j = 1, \ldots, m \tag{6}$$

where $P_i^g$ is a vector of correlation values between gene $i$ and other genes, and $P_{ij}^g$ is the Pearson correlation value between the $i$th gene and the $j$th gene. We set $p$-value $< 0.05$ to extract strongly correlated gene–gene pairs of gene $i$. The final correlation matrix was used as the adjacency matrix $A$ of the gene correlation network.

Next, we took the low-dimensional representations $E_{m \times d}^g$ encoded by deep neural networks as the feature matrix $E$ of nodes in the gene correlation network. $D$ was the degree matrix of the gene correlation network. For the VGAE, a 2-layer graph convolution network was defined as $GCN(E, A) = \tilde{A} ReLU(\tilde{A}_{m \times m} E W_0) W_1$, where $\tilde{A} = D^{-1/2} A D^{-1/2}$, and $W_0$ and $W_1$ are the learned weight matrices. The encoder of VGAE was defined as:

$$q(Z|E, A) = \prod_{i=1}^{m} \mathcal{N}\left(z_i | \mu_i, \ diag(\sigma_i^2)\right) \tag{7}$$

where $\mu_i$ is a mean vector from the matrix $\mu = GCN_\mu (E, A)$. Similarly, $\sigma_i$ is the variance and log $\sigma = GCN_\sigma (E, A)$; $Z$ is the representative matrix of the graph in low-dimensional space. $GCN_\mu (E, A)$ and $GCN_\sigma (E, A)$ share first-layer weight $W_0$.

The decoder reconstructed the network to generate a gene–gene association network by an inner product between latent variables:

$$p(A|Z) = \prod_{i=1}^{m} \prod_{j=1}^{m} p(A_{ij}|z_i,\ z_j), \text{ where } p\left(A_{ij} > 0|z_i,\ z_j\right) = sigmoid(z_i^T z_j) \qquad (8)$$

The goal of learning the VGAE was to optimize the variational lower bound $L$:

$$L = E_{q(Z|E,\ A)}[\log p(A|Z)] - KL[q(Z|E, A)||p(Z)] \qquad (9)$$

where $KL$ is the Kullback–Leibler divergence. According to our benchmark experiment, the learning rate was set as 0.01 and the number of iterations was 300 (S34A, S34C and S34D Fig).

For the cell–cell Pearson correlation matrix $P_{n \times n}^c$, we constructed a cell–cell association network using the GAE, similar to constructing a gene–gene association network.

## Calculating pathway activity scores and identifying gene modules

We normalized adjacency matrices of the gene–gene association network, cell–cell association network, and gene–cell association matrix by min-max normalization. These values represented the relative strength of gene–gene, cell–cell, and gene–cell associations. We spliced the adjacency matrix of the cell–cell association network with the gene–cell association matrix and the adjacency matrix of the gene–gene association network with the gene–cell association matrix by column (S35 Fig). The 2 spliced matrices were then merged by row to form a gene–cell association network. The $W$ is the column-normalized adjacency matrix of the gene–cell association network. For 1 pathway, we used the genes included in this pathway as restart nodes (called seeds) in the gene–cell association network. Subsequently, the RWR algorithm performed diffusion and iteration over the gene–cell association network using the seed nodes as starting nodes. The process was as follows.

$$p_{t+1} = (1 - r)Wp_t + rp^0 \qquad (10)$$

where $p^0$ is the initial probability vector, and only the seeds have nonzero values; $t$ is the number of iterations; and $r$ is the restart probability. Kohler and colleagues showed that $r$ had only a slight effect on the results of the RWR algorithm when it fluctuated between 0.1 and 0.9 [86]. This was also confirmed by our benchmark experiments for the $r$ values (S34E Fig). In this study, we set $r = 0.7$, which had relatively better performance (S34E Fig). We obtained the stationary probability vector by iterating repeatedly until the difference between $p_{t+1}$ and $p_t$ fell below $1 \times 10^{-6}$, i.e., $\Sigma|p_{t+1} - p_t| < 1 \times 10^{-6}$. When the iteration ended, the stationary probability values of each cell node obtained from the seed diffusion represented the proximity measure between each cell and pathway [87]. We further utilized permutation analysis, which randomly sampled the same number of genes as seeds, to adjust the probability values and take them as pathway activity scores:

$$PAS_{ij} = 1 - \frac{|p' \geq p_{ij}|}{N} \qquad (11)$$

where $PAS_{ij}$ is the pathway activity score of $i$th pathway in the $j$th cell, $p'$ is a vector of the perturbed stationary probability values, and $N$ is the number of perturbations. We set the number of perturbations to 100 (S34F Fig). || indicates the number of elements in the set.

We next used cells of the same phenotype as seeds to spontaneously identify cell phenotype–associated gene modules. For each gene, the RWR algorithm quantified the association strength with the seeds (cell phenotype), and the stationary probability value was used as the strength of association score. Subsequently, the permutations test, which randomly selected cells as seeds, estimated the statistical significance of each gene. Genes that were less than the significance threshold (the parameter could be set and the default value was 0.01 in scapGNN) comprised the cell phenotype–associated gene module. For genes significantly associated with multiple cell phenotypes, we normalized the association strength with the sum equal to 1 as the propensity of genes to be expressed between cell phenotypes and provided a visualization program for the network of cell phenotype–associated gene modules.

## Constructing the cell community network

An important feature of the network is the existence of the community or cluster structure, which is defined as a group of nodes having similar affiliations but different to rest of the network. In this study, we identify cell communities in the cell–cell association network by the community detection algorithms such as edge-betweenness, leading eigen, and louvain. Cells from the same community may tend to have similar biological functions. We fused cells belonging to the same phenotype into 1 cell community node and merged edges between 2 cell communities by averaging weights to generate a cell community network. In this way, it will be feasible to show the strength of the association between different cell types or cell communities. The edge-betweenness algorithm is implemented by the cluster_edge_betweenness function of the R package igraph v1.3.1. The leading eigen and louvain methods are also provided in our R package scapGNN.

## Integrating single-cell multi-omics data into pathway activity score matrix

For the single-cell transcriptome and epigenome from the same cells, we constructed gene–cell association networks of scRNA-seq and scATAC-seq data. We further merged the 2 gene–cell association networks into a new multi-omics gene–cell association network (S1 Fig). For edges shared by 2 networks, we merged the weights into 1 combined weight using Brown's method [88]. Brown's method was first used to combine multiple dependent statistical tests [88]. Many studies have used it for tasks such as multi-omics integration or for calculating pathway scores by combining the expression values of genes in the pathways [25,89]. Brown's method considered dependencies between datasets and thus provided more conservative estimates of significance for genes supported by multiple similar omics datasets [25]. The edge weights in the gene–cell association network served as indicators of significance between nodes, such as the strength of coexpression between genes, the level of gene expression in cells, and the similarity between cells. After integrating gene–cell association networks from different omics, the weights of the edges accounted for the overall covariation of the weights from different sources of evidence. Furthermore, we used the RWR algorithm to calculate the proximity between genes in the pathway and each cell as the multi-omics information-supported pathway activity scores and the proximity between cells with the same phenotype and each gene to identify cell phenotype–associated gene modules.

## Cell clustering performance evaluation

We benchmarked the cell clustering performance of scapGNN using 16 scRNA-seq datasets (S3 Table), which were preprocessed according to uniform standards. We downloaded the extensible markup language (XML) files of 396 pathways from the KEGG database and used the gene set contained in each pathway as a pathway term. We used the pathway activity score

matrix obtained from different single-cell pathway activity scoring methods as input to the sci-Path framework. The 10 state-of-the-art single-cell clustering methods (S4 Table) in the sci-Path framework were used to infer cell clustering from the pathway activity score matrix. The procedures and parameters organized by the sciPath framework were used [1]. In the 10 cell clustering methods, the Leiden algorithm of the Seurat process was used at 3 levels of resolution (0.5, 1, and 1.5). The top 2,000 highly variable genes were used for the cell clustering of gene level. The AUCell, Pagoda2, and UniPath were implemented using the scTPA [90] and UniPath package [14]. The default parameters were used. S2 Table provides guidance on the use of these methods. All pathways in the pathway activity score matrix were used and involved in cell clustering. The cell clustering performance was quantified using 3 accuracy indicators: ARI [91], NMI [92], and SW [93]. ARI and NMI were implemented using the sci-Path framework. SW was implemented using the R package cluster v2.1.3. The mean value of 3 accuracy indicator results of the 10 cell clustering methods was used (average ARI, average NMI, and average SW).

For cell clustering assessment of single-cell multi-omics data, the Cobolt v1.0.1 and GLUE v0.3.2 were used to integrate single-cell multi-omics data, and default parameters were used. The Leiden algorithm was used to cluster the cells and calculate the cell clustering indicators for each method at different levels of resolution (0.5, 1, and 1.5). The purity function of the R package funtimes v9.0 was used to calculate the purity of the clustering results.

## Evaluation of ability to identify pathways and gene modules

We used known cell marker gene sets to test the accuracy of scapGNN in identifying pathways such as the UniPath [14]. Marker genes for each cell type were collected from the CellMarker database, BioGPS, and Harmonizome and used as a gene set. Further, 460 marker gene sets corresponding to 460 cell types were collected by Smriti and colleagues [14] and used as the gold standard (S36 Fig). Both homogeneous and heterogeneous scRNA-seq datasets were used to calculate pathway activity scores (S1 Table). We calculated the proportion of cells that correctly detected cell type marker gene sets among the top 1 to 5 rankings in pathway activity scores. We used the KEGG pathway data from the C2 gene set of the molecular signatures database (MSigDB), including 186 biological pathways. Using these data, we identified the proportion of T and B cells that ranked the T- and B cell receptor signaling pathways within the top 1 to 5.

We further used marker genes of cell types as the gold standard. The ROC curve analysis was used to verify that scapGNN could identify cell phenotype–associated gene modules.

## Supporting information

**S1 Text. Supplementary methods.**
(DOCX)

**S1 Fig. The workflow of integrating single-cell multi-omics data by scapGNN.** First, the GNN model of scapGNN constructs a gene–cell association network for gene expression profiles of scRNA-seq data and gene activity matrix of scATAC-seq data, respectively. Second, Brown's method integrates 2 gene–cell association networks into a combined gene–cell association network. Finally, the RWR algorithm is used to calculate pathway activity scores and identify cell phenotype–associated gene modules with multi-omics information. GNN, graph neural network; RWR, random walk with restart; scATAC-seq, single-cell ATAC sequencing; scRNA-seq, single-cell RNA sequencing.
(PDF)

**S2 Fig. Nonlinear dimensional reduction visualization of pathway activity scores and difference analysis of the pathway activity.** tSNE visualizations of cell type data (**A**), cell subtype data (**B**), and time series data (**C**) based on pathway activity scores using the 4 pathway enrichment methods (AUCell, Pagoda2, UniPath, and scapGNN). The data underlying this figure can be found in S1 Data.
(PDF)

**S3 Fig. Bar graphs of 3 cell clustering accuracy indicators (ARI, NMI, and SW) that evaluated the cell clustering results of AUCell, Pagoda2, UniPath, and scapGNN in the 3 scRNA-seq datasets using the 10 cell clustering methods.** The data underlying this figure can be found in S7 Data. ARI, adjusted rand index; NMI, normalized mutual information; scRNA-seq, single-cell RNA sequencing; SW, silhouette width.
(PDF)

**S4 Fig.** Three starting points from the 0-h cell population in the time series dataset were selected to infer the pseudotime for AUCell (**A**), Pagoda2 (**B**), UniPath (**C**), and scapGNN (**D**). (**E**) Bar graphs of BCMI between pseudotimes inferred based on pathway activity scores and true cellular timestamps of the time series dataset. The data underlying this figure can be found in S7 Data.
(PDF)

**S5 Fig. Temporal-ordering analysis of AUCell, Pagoda2, UniPath, and scapGNN for the time series dataset following the strategy of UniPath.**
(PDF)

**S6 Fig. Cell community network of the time series dataset.** (**A**) A cell community network merged cell nodes of the same cell type. (**B**) Cell communities of the cell–cell association network were identified, and cell nodes of the same cell community were merged.
(PDF)

**S7 Fig. Application of traditional bulky RNA-seq pathway enrichment analysis methods to scRNA-seq data.** Box plot of average ARI, average NMI, and average SW of the pathway level from ssGSEA (**A**) and GSVA (**B**) using 10 state-of-the-art single-cell clustering methods on 16 scRNA-seq data sets. The data underlying this figure can be found in S1 Data. ARI, adjusted rand index; GSVA, gene set variation analysis; NMI, normalized mutual information; scRNA-seq, single-cell RNA sequencing; ssGSEA, single-sample gene set enrichment analysis; SW, silhouette width.
(PDF)

**S8 Fig. Performance of cell clustering was assessed according to detected gene numbers.** (**A**) Box plots of the average number of genes detected in cells for the high–gene number group datasets and the low–gene number group datasets. (**B**) Box plots of cell clustering accuracy indicators (ARI, NMI, and SW) for the 4 single-cell pathway activity scoring methods in the 2-group benchmark datasets. The data underlying this figure can be found in S7 Data. ARI, adjusted rand index; NMI, normalized mutual information; SW, silhouette width.
(PDF)

**S9 Fig. Box plot of cell clustering accuracy indicators (ARI and NMI) for AUCell, Pagoda2, UniPath, and scapGNN using the 10 cell clustering methods.** The data underlying this figure can be found in S7 Data.
(PDF)

**S10 Fig. Performance evaluation of scapGNN on scRNA-seq data with batch effects.** (**A**) UMAP visualizations included raw gene expression data, integrated gene expression data, and the pathway activity score matrix after data integration using Seurat v4. The graphs show the UMAP plots separated by cell type. (**B**) Bar plots of 3 cell clustering accuracy indicators (ARI, NMI, and SW) for using scapGNN, AUCell, Pagoda2, and UniPath on the integrated gene expression data. (**C**) Proportion of A549 cells that detected the corresponding correct cell type marker gene set in the top 1 to 5 of the pathway scores. The data underlying this figure can be found in S7 Data. ARI, adjusted rand index; NMI, normalized mutual information; scRNA-seq, single-cell RNA sequencing; SW, silhouette width; UMAP, Uniform Manifold Approximation and Projection.
(PDF)

**S11 Fig. Evaluation of performance on large-scale single-cell data.** (**A**) Cell clustering tSNE of AUCell, Pagoda2, UniPath, and scapGNN on the mouse cell atlas dataset. (**B**) Bar graph of cell clustering accuracy indicators (ARI, NMI, and purity). The data underlying this figure can be found in S7 Data. ARI, adjusted rand index; NMI, normalized mutual information; tSNE, t-distributed stochastic neighbor embedding.
(PDF)

**S12 Fig. Evaluation of scapGNN uniformity by ablation experiments on cell type dataset.** The data underlying this figure can be found in S8 Data.
(PDF)

**S13 Fig. Pathway score distribution and cell marker pathways in the T cell and B cell datasets.** (**A**) Score distribution of T-cell receptor signaling pathway activity for T cells with T-cell receptor signaling pathway ranked in the top 1 to 5. (**B**) Score distribution of B cell receptor signaling pathway activity for B cells with B cell receptor signaling pathway ranked in the top 1 to 5. Seurat was used to identify the top 5 B (**C**) and T (**D**) cell marker pathways. P_val_adj, adjusted *p*-value.
(PDF)

**S14 Fig. Stability analysis of the scapGNN identity marker pathway in the scRNA-seq data.** (**A**) Proportion of T cells with T-cell receptor signaling pathway or B cells with B cell receptor signaling pathway appeared in the top 5 enriched terms when T cells or B cells were grouped with epithelial cells in the T cell and B cell datasets. (**B**) Proportion of T cells with T-cell receptor signaling pathway or B cells with B cell receptor signaling pathway appeared in the top 5 enriched terms when T cells or B cells were grouped with monocytes in the scRNA-seq data of the PBMC multi-omics dataset. Untreated means that the original data were used. The data underlying this figure can be found in S2 Data.
(PDF)

**S15 Fig. Robustness analysis of the gene modules.** Using marker genes of the ESC as gold standards, ROC curves of ESC-associated gene module identified by scapGNN (**A**) or geno-map (**B**) in the ESC dataset with different strengths of dropout noise. The data underlying this figure can be found in S2 Data.
(PDF)

**S16 Fig. Functionally modular evaluation for cell phenotype–associated gene modules of activated stellate cells in the cell type dataset, eProg1b cells in the cell subtype dataset, and 36-h cells in the time series dataset.** The data underlying this figure can be found in S8 Data.
(PDF)

**S17 Fig. Robustness evaluation.** UMAP visualizations of the AUCell (**A**), Pagoda2 (**B**), and UniPath (**C**) on cell subtype datasets with different strengths of dropout noise. (**D**) AUC between the zero-valued rates and 3 cell clustering accuracy (ARI, NMI, and SW) quantification indicators on 16 scRNA-seq data sets. (**E**) Proportion of ESC cells with corresponding marker gene set in the top 1 to 5 pathway scores using the 4 pathway enrichment methods under different strengths of dropout noise on the ESC dataset. The data underlying this figure can be found in S3 Data. ARI, adjusted rand index; AUC, area under the recovery curve; ESC, embryonic stem cell; NMI, normalized mutual information; scRNA-seq, single-cell RNA sequencing; SW, silhouette width; UMAP, Uniform Manifold Approximation and Projection. (PDF)

**S18 Fig. Evaluation of the stability of scapGNN in scATAC-seq data of the mouse cortical brain dataset for pathway activity scoring.** (**A**) Proportion of astrocytes with corresponding marker gene set in the top 1 to 5 pathway scores using the 4 pathway enrichment methods under different strengths of dropout noise. (**B**) The proportion of astrocytes with the marker gene set of astrocytes appeared in the top 5 enriched terms when astrocytes were grouped with layer 6b (L6b) or vasoactive intestinal polypeptide (Vip) cells. Untreated means that the original data were used. The data underlying this figure can be found in S4 Data. (PDF)

**S19 Fig. Pathway score distribution and cell marker pathways in the PBMC dataset.** (**A**) Score distribution of T-cell receptor signaling pathway activity for T cells with T-cell receptor signaling pathway ranked in the top 1 to 5. (**B**) Seurat was used to identify the top 5 T-cell marker pathways. P_val_adj: adjusted *p*-value. (PDF)

**S20 Fig. Stability analysis of the scapGNN identity marker pathway in the scATAC-seq data.** The proportion of T cells with T-cell receptor signaling pathway appeared in the top 5 enriched terms when T cells were grouped with B cells or monocytes. Untreated means that the raw data were used. The data underlying this figure can be found in S4 Data. (PDF)

**S21 Fig. Performance of scapGNN for cell clustering in single-cell multi-omics data integration.** UMAP visualization of the gene expression matrix from scRNA-seq, the gene activity score matrix from scATAC-seq, and the pathway activity score matrix from single-cell multi-omics integration of scapGNN for the mouse brain cortex dataset (**A**), mouse skin dataset (**B**), and PBMC multi-omics dataset (**C**). (**D**) Bar graph of the 3 cell clustering accuracy indicators for scapGNN, and the state-of-the-art single-cell multi-omics integration methods on the mouse brain cortex dataset. (**E**) Bar graph of the cell clustering accuracy indicator for single-cell transcriptomic and single-cell epigenomic data from the mouse brain cortex, mouse skin, and PBMC multi-omics datasets. The data underlying this figure can be found in S5 Data. PBMC, peripheral blood mononuclear cell; scATAC-seq, single-cell ATAC sequencing; scRNA-seq, single-cell RNA sequencing; UMAP, Uniform Manifold Approximation and Projection. (PDF)

**S22 Fig. Performance of scapGNN for pseudotime inference in single-cell multi-omics data integration.** Latent embedding of scDART and visualization using PCA on the mouse skin dataset. Cells are colored with cell type (**A**) and inferred pseudotime (**B**). Red arrows are TAC populations pointing to the medulla, IRS, and hair shaft cuticle/cortex cells. The data underlying this figure can be found in S5 Data. IRS, inner root sheath; PCA, principal

component analysis; TAC, transit-amplifying cell.
(PDF)

**S23 Fig. scapGNN was capable of accurately identifying active pathways in single-cell multi-omics data.** (**A**) Proportion of cells with T- or B cell receptor signaling pathway in the top 5 pathways for the PBMC multi-omics dataset. (**B**) ROC curves of gene modules of T and B cells in the PBMC multi-omics dataset with T- and B cell receptor signaling pathway genes as gold standards. (**C**) Proportion of cells that detected the correct marker gene sets using scapGNN in the top 1 to 5 of the single-cell multi-omics–supported pathway activity scores. (**D**) ROC curves of gene modules of the endothelial cells and GM12878 with marker gene sets for each cell type as gold standards. The endothelial cells are from the mouse skin dataset, and GM12878 is from the GM12878 dataset. The data underlying this figure can be found in S5 Data.
(PDF)

**S24 Fig. Stability analysis of the scapGNN in identifying the marker pathway in the single-cell multi-omics integration.** The proportion of T cells with T-cell receptor signaling pathway appeared in the top 5 enriched terms when T cells were grouped with monocytes. Untreated means that the raw data were used. The data underlying this figure can be found in S5 Data.
(PDF)

**S25 Fig. Developmental process–dependent and hedgehog signaling pathways in TACs.** Violin plots of Gli2 (**A**) and Gli3 (**B**) activity in scRNA-seq and scATAC-seq data from the mouse skin dataset. (**C**) UMAP plots colored by the single-cell multi-omics supported pathway activity scores of the hedgehog signaling pathway. (**D**) Network of cell phenotype–associated gene modules during the proliferation of TACs in hair follicle tissue. Red boxes mark the locations of Gli2 and Gli3. The data underlying this figure can be found in S5 Data. scATAC-seq, single-cell ATAC sequencing; scRNA-seq, single-cell RNA sequencing; TAC, transit-amplifying cell; UMAP, Uniform Manifold Approximation and Projection.
(PDF)

**S26 Fig. Cell differentiation trajectory analysis of mouse early embryo development using pathway activity scores from scapGNN, AUCell, Pagoda2, and UniPath.** The data underlying this figure can be found in S6 Data.
(PDF)

**S27 Fig. An extension of the application in reproductive biology.** (**A**) Gene association network of oxidative phosphorylation pathway in the early pachytene and steps 3–4 spermatids. The node size reflects the importance of the genes, and the width of the edges between nodes indicates the strength of association between genes in the network. (**B**) Dynamic expression of the PPAR signaling pathway during early embryonic development as counted by the EmExplorer database. (**C**) Cell community association network of human testis data. The width of the edges indicates the strength of association between cell communities. (**D**) Network of cell type–associated gene modules between Sertoli cells and spermatogenic cells in the human testis dataset. In the network, the sector area of a node indicates the strength of association between a gene and a cell phenotype, and the width of the edges between nodes indicates the strength of association between genes. The data underlying this figure can be found in S6 Data.
(PDF)

**S28 Fig. Box plots of expression values of genes associated with *Fabp9* in each stage of spermatogenesis.** *Rbakdn* (**A**) and *Spata33* (**B**) were the 2 genes with the largest association scores

with *Fabp9. Cypt12* (**C**) and *Hmgb4* (**D**), 2 genes with smaller association scores with *Fabp9*, served as controls. The data underlying this figure can be found in S6 Data.
(PDF)

**S29 Fig. Application of scapGNN to COVID-19 data.** (**A**) UMAP visualizations of the scapGNN-derived pathway activity matrix for healthy controls and patients with COVID-19. (**B**) Gene association network of thyroid hormone synthesis in patients with COVID-19. The node size reflects the importance of the genes, and the width of the edges between nodes indicates the strength of association between genes in the network. (**C**) Network of cell pheno-type–associated gene modules in healthy controls, patients with COVID-19, and those with IAV. In the network, the sector area of a node indicates the strength of association between a gene and a cell phenotype, and the width of the edges between nodes indicates the strength of association between genes. The data underlying this figure can be found in S8 Data. COVID-19, coronavirus disease 2019; IAV, influenza A virus; UMAP, Uniform Manifold Approxima-tion and Projection.
(PDF)

**S30 Fig. Heatmap of the activity scores of the B cell–associated differential pathway between the healthy controls and patients with COVID-19.** The annotations in the columns of the heat map indicate the phenotype to which the cell belongs. The row annotations indicate that the pathway is significantly different in the phenotype of the cells. The data underlying this figure can be found in S8 Data.
(PDF)

**S31 Fig. Evaluation of runtime.** (**A**) Runtime of the GNN module of scapGNN in different cell-scale scRNA-seq datasets. (**B**) Time (min) of pathway activity scores for scapGNN in dif-ferent cell-scale scRNA-seq datasets. (**C**) Time (s) of pathway activity scores for AUCell, Pagoda2, and UniPath in different cell-scale scRNA-seq datasets. The data underlying this fig-ure can be found in S8 Data.
(PDF)

**S32 Fig. Effect of increasing the number of genes in heterogeneous data on the perfor-mance of scapGNN.** Genes, initially 2,000 highly variable genes and then increasing in count, were used to calculate individual cell pathway activity scores for the cell type dataset. (**A**) Cell clustering indicators based on pathway activity scores at different gene numbers. (**B**) Propor-tion of endothelial cells with the corresponding marker gene set in the top 5 for different gene counts. The data underlying this figure can be found in S8 Data.
(PDF)

**S33 Fig. Effect of increasing the number of genes in homogeneous data on the performance of scapGNN.** Genes, initially 2,000 highly variable genes and then increasing in count, were used to calculate individual cell pathway activity scores. (**A**) Proportion of T cells with the cor-responding T-cell receptor signaling pathway in the top 5 for different gene counts. (**B**) Scatter plots show the relationship between the genes in the T-cell receptor signaling pathway and the 8,000 highly variant genes that were screened. (**C**) Proportion of K562 cells with the corre-sponding marker gene set in the top 5 for different gene counts. (**D**) Scatter plots showing the relationship between the genes in the marker gene set of the K562 cells and the 8,000 highly variant genes that were screened. The data underlying this figure can be found in S8 Data.
(PDF)

**S34 Fig. Hyperparameter evaluation of the scapGNN using the cell type dataset.** Cell clus-tering indicators based on pathway activity scores and the proportion of endothelial cells with

the corresponding marker gene set in the top 5 at different learning rates for DNNAE (**A**), number of iterations for DNNAE (**B**), learning rate for GAE (**C**), number of iterations for GAE (**D**), restart probability values (**E**), and number of perturbations (**F**). The data underlying this figure can be found in S8 Data.
(PDF)

**S35 Fig. Flowchart of constructing gene–cell association networks by integrating adjacency matrices.** C, adjacency matrix of the cell–cell association network; K, gene–cell association matrix; G, adjacency matrix for the gene–gene association network.
(PDF)

**S36 Fig. Distribution of the number of genes in the cell type marker gene set.** (**A**) Number of genes contained in marker gene sets. (**B**) Number of genes contained in marker gene sets for the cell types used in evaluation analysis. The data underlying this figure can be found in S8 Data.
(PDF)

**S1 Table. scRNA-seq datasets for the scapGNN application.**
(DOCX)

**S2 Table. State-of-the-art methods for comparing performance and manuals.**
(DOCX)

**S3 Table. Benchmark datasets for the performance evaluation of cell clustering.**
(DOCX)

**S4 Table. Single-cell clustering methods.**
(DOCX)

**S5 Table. Omics datasets for the evaluation of pathway and gene module identification.**
(DOCX)

**S6 Table. Pathways with significant differences in certain types of cells only in integrated multi-omics data of scRNA-seq and scATAC-seq from adult mouse brain**
(DOCX)

**S1 Data. Supporting data for Fig 2, S2 and S7 Figs.**
(XLSX)

**S2 Data. Supporting data for Fig 3, S14 and S15 Figs.**
(XLSX)

**S3 Data. Supporting data for Fig 4 and S17 Fig.**
(XLSX)

**S4 Data. Supporting data for Fig 5, S18 and S20 Figs.**
(XLSX)

**S5 Data. Supporting data for Fig 6, S21, S22, S23, S24 and S25 Figs.**
(XLSX)

**S6 Data. Supporting data for Fig 7, S26, S27 and S28 Figs.**
(XLSX)

**S7 Data. Supporting data for S3, S4, S8, S9, S10 and S11 Figs.**
(XLSX)

**S8 Data. Supporting data for S12, S16, S29, S30, S31, S32, S33, S34 and S36 Figs.**
(XLSX)

## Author Contributions

**Conceptualization:** Hui Zhu, Yan Li, Xuejiang Guo.

**Data curation:** Xudong Han, Bing Wang, Hui Zhu, Yan Li, Xuejiang Guo.

**Formal analysis:** Chenghao Situ, Yaling Qi.

**Funding acquisition:** Hui Zhu, Yan Li, Xuejiang Guo.

**Investigation:** Xudong Han, Bing Wang, Chenghao Situ, Yaling Qi, Hui Zhu, Yan Li, Xuejiang Guo.

**Project administration:** Hui Zhu, Yan Li, Xuejiang Guo.

**Resources:** Xuejiang Guo.

**Software:** Xudong Han, Xuejiang Guo.

**Validation:** Xudong Han.

**Visualization:** Xudong Han, Bing Wang.

**Writing – original draft:** Xudong Han.

**Writing – review & editing:** Hui Zhu, Yan Li, Xuejiang Guo.

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
