## [Editor Report · Decision Letter 0]

28 Feb 2023

Dear Dr Guo, 

Thank you for submitting your manuscript entitled "scapGNN: a graph neural network-based framework for active pathway and gene module inference from single-cell multi-omics data" for consideration as a Methods and Resources Article by PLOS Biology. Please accept my sincere apologies for the delay in getting back to you as we consulted with an academic editor about your submission.

Your manuscript has now been evaluated by the PLOS Biology editorial staff, as well as by an academic editor with relevant expertise, and I am writing to let you know that we would like to send your submission out for external peer review.

Once your full submission is complete, your paper will undergo a series of checks in preparation for peer review. After your manuscript has passed the checks it will be sent out for review. To provide the metadata for your submission, please Login to Editorial Manager (https://www.editorialmanager.com/pbiology) within two working days, i.e. by Mar 02 2023 11:59PM.

Kind regards,

Richard

Richard Hodge, PhD

Associate Editor, PLOS Biology

rhodge@plos.org

PLOS

---

## [Decision Letter · Decision Letter 1]

25 Apr 2023

Dear Dr Guo,

Thank you for your patience while your manuscript "scapGNN: a graph neural network-based framework for active pathway and gene module inference from single-cell multi-omics data" was peer-reviewed at PLOS Biology. Please accept my sincere apologies for the delays that you have experienced during the peer review process. Your manuscript has now been evaluated by the PLOS Biology editors, an Academic Editor with relevant expertise, and by two independent reviewers. 

In light of the reviews, which you will find at the end of this email, we would like to invite you to revise the work to thoroughly address the reviewers' reports.

As you will see, the reviewers are positive about your manuscript and note that the method is timely and will be of use to the field. However, they raise overlapping concerns with the strength of the benchmarking experiments and ask that newer techniques are used for comparisons, as well as using scapGNN on multi-omics datasets. We feel that the addition of these benchmarking experiments would be essential to consider a revised version of your manuscript. In addition, the reviewers ask that additional discussions about how scapGNN will advance our understanding of single cell biology are included, as well as providing a broader contextualization and discussion of previous methods.

Given the extent of revision needed, we cannot make a decision about publication until we have seen the revised manuscript and your response to the reviewers' comments. Your revised manuscript is likely to be sent for further evaluation by all or a subset of the reviewers.

**IMPORTANT - SUBMITTING YOUR REVISION**

*Re-submission Checklist*

*Published Peer Review*

*PLOS Data Policy*

*Blot and Gel Data Policy*

Sincerely,

Richard

Richard Hodge, PhD

Associate Editor, PLOS Biology

rhodge@plos.org

REVIEWS:

Reviewer #1: In this manuscript, the authors propose present scapGNN, a graph neural network (GNN)-based framework that calculates pathway activity scores, identifies cell phenotype-associated gene modules, and integrates multi-omics data at single-cell resolution. There exist a few scientific and presentation issues to be fixed.

Scientific issues:

1. Does selecting highly variable genes in data preprocessing eliminate some genes from the pathway and thus make the calculated activity score inaccurate, especially in homogeneous data?

2. Since the application scenario is different from Kohler's study, please do experiments to show that different choices of r do not have a large impact on the result of the RWR algorithm.

3. Whether the three networks (gene-gene association network, cell-cell association network, gene-cell association network) on which the random walk is based can be compared at the same level?

4. What’s the significance of constructing the cell community network? What can users use it to do? Please prove to construct it is significant.

5. Why use multi-omics to explain mechanisms at the pathway level? Can't we effectively analyze pathways using only scRNA-seq data? There are not enough experiments to show that using multi-omics data can provide a better improvement than single-omics data. The proof based on only one set of data is not convincing. Moreover, the only two cell alignment methods (Cobolt and GLUE) which scapGNN is compared with in terms of cell clustering performance are not enough. There are many methods that can do cell alignments, such as Seuratv4, MOFA2.0, TotalVI, and Deepmaps. And due to these methods, all using deep learning models, these methods can all be compared by means of adjusting hyperparameters, which makes it more convincing that scapGNN is better than others. Besides the experiments done on scRNA-seq datasets, should be also used on multi-omics datasets, such as the ability to compute pseudo time, the ability to find subclusters, the ability to find the true active pathways on homogeneous and heterogeneous data, and the ability to identify cell phenotype-associated gene modules. Through these experiments, you can prove that the scapGNN model based on multi-omics data is better than the scapGNN model and other methods based on single-omics data.

6. The title of this article is “scapGNN: a graph neural network-based framework for active pathway and gene module inference from single-cell multi-omics data”. However, most of the benchmark experiments in this paper are based on scRNA-seq data, and the only case study is also based on scRNA-seq data. Please supplement the benchmark experiment and case study based on multi-omics data suffice to show that this article is designed for multi-omics data.

7. Are all the experiments in this paper based on the same set of parameters of scapGNN and other benchmark methods? Do other benchmark methods use their default parameters? Please provide a parameter description. 

8. Please compare these methods’ performance on a grid of hyperparameters to prove this method outperforms other methods not just on a particular set of parameters and compare the robustness of the methods to the parameters.

9. Prove that LTMG helps improve model performance in ablation experiments

10. Version targeting single-cell pathway enrichment analysis (ssGSEA) are available in GSEA. Why doesn’t the user choose ssGSEA to get single-cell pathway enrichment? Please explain if there are any shortcomings in its method and do an experiment to compare with it.

11. Please explain the significance of constructing a pathway activity matrix on a single-cell level. Can use it to replace a single-cell expression matrix on single-cell data analysis? If so, please prove it will have a significantly better performance on downstream analysis such as cell clustering, and trajectory inference, than based on a single-cell expression matrix.

12. In S12 Fig, did the experiments on T cell +epithelial cell and B cell +epithelial cell eliminate the batch effect? For the experiment in S12 Fig, why not use a PBMC dataset? The T cell +epithelial cell doesn’t have enough heterogeneity. Please do this experiment on a PBMC dataset that contains multiple cell types and prove it will also have a better performance based on a multi-omics dataset.

13. On line 288, why use the biomarkers of cell clusters identified by Seurat v4 as the gold standard and not cell type marker genes in the databases? If the datasets don’t have cell type marker genes information in the databases, why not use datasets whose cell types have recognized marker information? This is also for robustness experiments.

14. Please compare with other software that do cell phenotype-related gene modules. This is also for robustness experiments.

15. What significant conclusion can the user get by identifying the differentially active pathways between the cell types? Or what can we do by using the differentially active pathways between the cell types?

16. Other software uses cell phenotype-related gene modules to do cell type-related pathway enrichment or study the regulatory mechanisms related to cell type or others. The case study didn’t show what significant results users can get by using cell type-related gene modules. 

17. In the case study, what biological significance do the genes that were found associated with multiple cell types represent? What research can users do with them?

18. In line 414, “The cell phenotype-associated gene networks discovered the regulatory mechanisms between cell types” didn’t be proven. How can we get regulatory mechanisms from the gene-gene edge in the network of cell phenotype-associated gene modules? Can the edges between nodes give the user information on the regulatory relationship between genes? Please improve it. 

Presentation issues:

1. W^l in equations (2) and (3) should be represented by different symbols.

2. The second item in equation(5) is problematic.

3. On page 28, “We obtain the stationary probability vector by iterating repeatedly until the difference between p_(t+1) and p_t falls below 1×10^(-6).” How to evaluate the difference between p_(t+1) and p_t by a single value.

4. Do the edges between genes in the network of cell type-associated gene modules come from the gene-gene association network, and what does the width of an edge represent?

5. Please explain clearly how to use single-cell pathway enrichment methods with cell clustering methods. Is it replacing the expression matrix with the pathway activity matrix?

6. Please provide the formula for calculating BCMI.

7. Please explain how Brown’s method works and why can use to integrate gene-gene, cell-cell, and gene-cell networks.

8. S8A does not significantly reflect that scapGNN can further eliminate the batch effect, especially for H1795 and H228 cell types. Please provide their separate UMAP plots or gray out the color of other types of cell points.

9. On line 281, “we separately set the cells of the four cell types (K562, A549, GM12878, and ESC)” have a problem. Are GM12878 and ESC not heterogeneous datasets? Please provide detailed information about the cell types in the datasets. 

10. In Fig6, there should be UMAP visualization for scRNA-seq and scATAC-seq data.

11. The legend of Fig7.E is wrong.

12. In line 413, S20C Fig can’t show Sertoli cells were tightly connected to spermatogenic cells.

13. In line 416, PSAP is not in gene modules of spermatogenic cells but in Spermatic cells.

14. Fig 7A can’t clearly show scapGNN can better reconstruct the process of spermatogenesis and early embryonic development. Please provide the development sort of cell type labels and the pseudo time inferred from Molocle3 on the plot. And please draw red arrows for other methods.

Reviewer #2: In the manuscript, the authors have proposed scapGNN, a graph neural network (GNN) -based framework that calculates pathway activity scores, identifies cell phenotype-associated gene modules, and integrates multi-omics data at single-cell resolution. The paper is overall well-written and easy to go through. The figures are also well-prepared. Although the problem is very important and timely, the paper has some issues that need to be addressed.

1. Abstract: In the abstract, lots of unnecessary sentences are added for what scapGNN does. However, it is not clear how it works. Please write as least two sentences on the novelty and major breakthrough scapGNN will bring to the data analysis pipeline. Please add a sentence on the improvement of quantitative performance of scapGNN with respect to other existing methods. 

2. Introduction: Many recent methods such as Refs.1-5 published in the last 2 years on the topic of deep genomic data analysis, gene co-expression network analysis and multi-omic data integration were not discussed. Please discuss the techniques (specially those developed on deep learning) and what are their limitations when compared to scapGNN.

3. Benchmarking: I see that the benchmarking is done mostly with old techniques. Please compare your technique with at least 2 techniques that were published in last year or this year. 

4. Discussion: The discussion section needs to be more rich. Specially, I would expect the authors to add some discussion on how scapGNN will enrich our understanding of single cell biology in case of new diseases such as COVID.

References:

1. Islam, M.T., Xing, L. Cartography of Genomic Interactions Enables Deep Analysis of Single-Cell Expression Data. Nat Commun 14, 679 (2023).

2. Gao, C., Liu, J., Kriebel, A.R. et al. Iterative single-cell multi-omic integration using online learning. Nat Biotechnol 39, 1000-1007 (2021).

3. Islam, M.T., Wang, JY., Ren, H. et al. Leveraging data-driven self-consistency for high-fidelity gene expression recovery. Nat Commun 13, 7142 (2022). 

4. Korsunsky, I., Millard, N., Fan, J. et al. Fast, sensitive and accurate integration of single-cell data with Harmony. Nat Methods 16, 1289-1296 (2019).

5. https://www.biorxiv.org/content/10.1101/2023.03.10.532124v1.abstract

---

## [Decision Letter · Decision Letter 2]

29 Aug 2023

Dear Dr Guo,

Thank you for your patience while we considered your revised manuscript "scapGNN: a graph neural network-based framework for active pathway and gene module inference from single-cell multi-omics data" for consideration as a Methods and Resources at PLOS Biology. Please accept my apologies for the delays that you have experienced during this round of the peer review process. Your revised study has now been evaluated by the PLOS Biology editors, the Academic Editor and the original reviewers. Please note that I have also provided some specific comments from the Academic Editor below the full reviews beneath my signature.

As you will see, both reviewers think that the manuscript is improved but they have some remaining concerns. Specifically, the reviewers raise concerns with the overall strength of the validation of scapGNN on multi-omics datasets and the benchmarking analyses (e.g. genomap). After discussions with the Academic Editor, we will not make the testing of additional multi-omics datasets essential for the revision and these comments could be addressed by textual edits where the utility of the GNN algorithm in analyzing these datasets is deemphasized. In addition, Reviewer #2 also comments upon the quality of the writing in the manuscript and we strongly encourage you to enlist the services of a professional editing service or a English-speaking colleague to improve the language. 

In light of the reviews, we are pleased to offer you the opportunity to address the comments from the reviewers in a revision that we anticipate should not take you very long. We will then assess your revised manuscript and your response to the reviewers' comments with our Academic Editor aiming to avoid further rounds of peer-review, although might need to consult with the reviewers, depending on the nature of the revisions.

In addition, I would be grateful if you could please address the following editorial and data-related requests that I have provided below (A-E):

(A) You may be aware of the PLOS Data Policy, which requires that all data be made available without restriction: http://journals.plos.org/plosbiology/s/data-availability. For more information, please also see this editorial: http://dx.doi.org/10.1371/journal.pbio.1001797

-Supplementary files (e.g., excel). Please ensure that all data files are uploaded as 'Supporting Information' and are invariably referred to (in the manuscript, figure legends, and the Description field when uploading your files) using the following format verbatim: S1 Data, S2 Data, etc. Multiple panels of a single or even several figures can be included as multiple sheets in one excel file that is saved using exactly the following convention: S1_Data.xlsx (using an underscore).

-Deposition in a publicly available repository. Please also provide the accession code or a reviewer link so that we may view your data before publication. 

Figure 2A-D, 3A-E, 4A-D, 5A-F, 6A-D, 7A-E, S2A-C, S3, S4A-F, S5, S7, S8, S9, S10A-C, S11A-B, S12, S14, S15, S16, S17A-E, S18A-B, S19, S20, S21A-D, S22, S23A-D, S24, S25A-D, S26, S27B, S27D, S28A-D, S29A-C, S31, S32A-B, S33A-D, S34A-F, S36A-B 

(B) Please also ensure that each of the relevant figure legends in your manuscript include information on *WHERE THE UNDERLYING DATA CAN BE FOUND*, and ensure your supplemental data file/s has a legend.

(C) Thank you for already providing the underlying code in Github (https://github.com/GaoLabXDU/sciPath). At this time, we ask that you please attach the Github deposition to a repository with long-term maintenance (Zenodo, https://zenodo.org/) and assign the deposition a DOI. 

(D) Please ensure that your Data Statement in the submission system accurately describes where your data can be found and is in final format, as it will be published as written there. This includes referencing where the code can be found and providing the DOI of the deposition. 

(E) Please also provide a blurb which (if accepted) will be included in our weekly and monthly Electronic Table of Contents, sent out to readers of PLOS Biology, and may be used to promote your article in social media. The blurb should be about 30-40 words long and is subject to editorial changes. It should, without exaggeration, entice people to read your manuscript. It should not be redundant with the title and should not contain acronyms or abbreviations. For examples, view our author guidelines: https://journals.plos.org/plosbiology/s/revising-your-manuscript#loc-blurb

We expect to receive your revised manuscript within 2 months. Please email us (plosbiology@plos.org) if you have any questions or concerns, or would like to request an extension. 

**IMPORTANT - SUBMITTING YOUR REVISION**

*Resubmission Checklist*

*Published Peer Review*

*PLOS Data Policy*

*Blot and Gel Data Policy*

Sincerely,

Richard

Richard Hodge, PhD

rhodge@plos.org

REVIEWS:

Reviewer #1: The author has given a very detailed replay to my comments but there are still two problems that need to be solved.

1. For Comment 5, two more multi-omics datasets used for evaluation should be added. There re only 3 multi-omics datasets and scapGNN has no advantage on one of the datasets. And, the scapGNN's cell clustering performance comparison between single omics and multi-omics cannot be done only through UMAP visualization, but also on cell clustering accuracy indicators.

2. For Comment 12, why not use a PBMC dataset containing three and more cell types? 

Reviewer #2: I will focus on the authors' response to my comments. 

1. The authors added in the manuscript: "Genomap simply

calculates the average gene activity values for each gene in the specified cell type as the strength of

association with the cell type. And genomap cannot provide significance-level p-values to screen

for those genes that make up a meaningful gene set." The first sentence is scientifically incorrect. The second sentence is grammatically not good. For the first sentence, I would recommend the authors to read the referred paper carefully. For second sentence, I would recommend them taking help from grammar-correcting tools. 

2. The authors discussed a few deep learning based techniques I referred in the response. However, I don't find them in the main manuscript. 

3. In Fig. S15, the authors compared the performance of genomap with scapGNN. However, the results look impractical. I would recommend the authors follow the example codes of genomap properly (specially the data standardization) to recreate the results. Please follow example 7 in https://github.com/xinglab-ai/genomap.

*COMMENTS FROM THE ACADEMIC EDITOR*

I agree with Reviewer #2 about the overall poor writing notably in the parts rewritten that resulted from the revision and the need to better evaluate genomap performance.

Even several of the revisions to Reviewer #1, although the additional analysis raised by the reviewer were performed, were not properly articulated in the new ms. (e.g. Point 1 on heterogenous and homogenous “marker” genes ; but also other points.)

So the author should put effort in the writing (English and precise articulation)

Specific points:

(i) I agree with Reviewer #1 that visual UMAP analysis is not solid way for performance comparison -this point can be addressed.

(ii) What is questionable is the request for testing more muti-omics data sets. For single-cell analysis these are rare and of questionable quality – I am not aware of any good ones. Reviewer #2 had suggested to deemphasize the multi-omics aspect and that is more appropriate. The authors should adjust the text and discussion accordingly. Eg pointing to these challenges of getting good quality multi-omics datasets – which are only now appearing on he “market” and present the multiomics as additional possible filed of applying their algorithm.

(iii) Similarly, PBMC sets with more cell types is readily available – but I wouldn’t make this essential since this is already a complex manuscript.

---

## [Editor Report · Decision Letter 3]

7 Oct 2023

Dear Dr Guo,

Thank you for the submission of your revised Methods and Resources Article "scapGNN: a graph neural network–based framework for active pathway and gene module inference from single-cell multi-omics data" for publication in PLOS Biology. On behalf of my colleagues and the Academic Editor, Sui Huang, I am pleased to say that we can accept your manuscript for publication, provided you address any remaining formatting and reporting issues. These will be detailed in an email you should receive within 2-3 business days from our colleagues in the journal operations team; no action is required from you until then. Please note that we will not be able to formally accept your manuscript and schedule it for publication until you have completed any requested changes.

PRESS

Best wishes, 

Richard

Richard Hodge, PhD

rhodge@plos.org

PLOS
